# Correlates of protection for booster doses of the SARS-CoV-2 vaccine BNT162b2

Tomer Hertz [1,2,3,18,19] ✉, Shlomia Levy[1,2,18], Daniel Ostrovsky[4,18], Hanna Oppenheimer[1,2,18], Shosh Zismanov[1,2,18], Alona Kuzmina[1,18], Lilach M. Friedman [1,2], Sanja Trifkovic [5], David Brice [6], Lin Chun-Yang[6], Liel Cohen-Lavi[1,2], Yonat Shemer-Avni[1,7], Merav Cohen-Lahav [8], Doron Amichay[9], Ayelet Keren-Naus[1,7], Olga Voloshin[7], Gabriel Weber[10,11], Ronza Najjar-Debbiny[10,11], Bibiana Chazan[11,12], Maureen A. McGargill [6], Richard Webby [5], Michal Chowers[13,14], Lena Novack[4], Victor Novack[4], Ran Taube [1,19] ✉, Lior Nesher [15,19] ✉ & Orly Weinstein[16,17]

Vaccination, especially with multiple doses, provides substantial population-level protection against COVID-19, but emerging variants of concern (VOC) and waning immunity represent significant risks at the individual level. Here we identify correlates of protection (COP) in a multicenter prospective study following 607 healthy individuals who received three doses of the Pfizer-BNT162b2 vaccine approximately six months prior to enrollment. We compared 242 individuals who received a fourth dose to 365 who did not. Within 90 days of enrollment, 239 individuals contracted COVID-19, 45% of the 3-dose group and 30% of the four-dose group. The fourth dose elicited a significant rise in antibody binding and neutralizing titers against multiple VOCs reducing the risk of symptomatic infection by 37% [95%CI, 15%-54%]. However, a group of individuals, characterized by low baseline titers of binding antibodies, remained susceptible to infection despite significantly increased neutralizing antibody titers upon boosting. A combination of reduced IgG levels to RBD mutants and reduced VOC-recognizing IgA antibodies represented the strongest COP in both the 3-dose group (HR = 6.34, $p$ = 0.008) and four-dose group (HR = 8.14, $p$ = 0.018). We validated our findings in an independent second cohort. In summary combination IgA and IgG baseline binding antibody levels may identify individuals most at risk from future infections.

Covid-19 is a disease caused by SARS-CoV-2 and has been driving a worldwide pandemic for the past three years. The pandemic has a broad spectrum of effects ranging from increased patient morbidity and mortality to impacting the global economy[1]. The rapid development of vaccines is the primary determinant in reducing this impact. The mRNA vaccines minimized infectivity and reduced hospitalizations, severe disease, and death[2]. However, not enough is known regarding the duration of protection or the schedule of boosting

required[3] SARS-CoV-2 has rapidly evolved, and variants of concern (VOC) have swept the world every few months, with the omicron variant and its sub-variants currently being the most common VOCs.

The Pfizer-BioNTech mRNA vaccine was initially administered in two doses 21 days apart. However, the third dose of Pfizer-BioNTech was approved in Israel in August 2021 (6 months following the initial doses) and subsequently worldwide to combat the Delta variant and waning of vaccine-elicited antibody responses. Multiple studies

reported that the third dose was very effective at inducing high neutralizing antibody levels[4,5] and preventing disease development and hospitalization complications[6]. Towards the end of 2021, the Omicron BA.1 variant which harbors up to 59 mutations throughout its genome, with 32 positioned within the spike and 15 within the receptor binding domain (RBD)[7–9] rapidly spread worldwide. BA.1. On January 2, 2022, the Israeli health ministry recommended the fourth dose of the Pfizer-BioNTech mRNA vaccine for immunocompromised groups. A fourth dose was also offered to healthcare providers (HCP) and people older than 60 years[10]. Epidemiological studies on the fourth dose out of Israel demonstrated its effectiveness in reducing infection rates; however, these studies evaluated persons over 60[11,12] or with a median age of 60[13].

Correlates of protection are immune markers that can be used to predict vaccine efficacy against infection or disease after vaccination[14–17]. Neutralizing antibodies or binding antibodies have been established as a correlate of protection for vaccines against many viral diseases[15]. More specifically, recent studies have shown that neutralizing antibody titers and IgG binding titers to the SARS-Cov-2 spike protein are correlates of protection from symptomatic infection following vaccination with mRNA vaccines and the ChadOx Astrazeneca vaccine[16,18–21]. Additional studies have highlighted the role of cellular responses as correlates of protection and in reducing disease severity[22–25].

Here we report an interim analysis of the Clalit HCPs Booster study—a multicenter prospective trial in healthcare providers with increased risk of SARS-CoV-2 infection designed to identify novel correlates of protection (COP) for booster doses of the Pfizer-BioNTech vaccine. We hypothesized that immune history to SARS-CoV-2, including both the Wuhan wildtype strain and multiple variants of concern, would be associated with infection risk with the Omicron VOC. Our analysis identified multiple correlates of protection that included both IgG and IgA immune markers. We further showed that these markers can identify a subpopulation of individuals with low SARS-CoV-2 baseline immune history that were at high risk of infection despite responding to an additional booster dose. Our findings suggest combinations of IgA and IgG binding antibody baseline immune markers provide improved correlates of protection.

## Results

We enrolled 639 HCP from four medical centers between January 6 and February 9, 2022. Of the 639 enrolled participants, 32 were excluded (see methods), and 607 individuals were included in the final analysis. All participants previously received a primary vaccine series of two doses and a third dose six months later. The median number of days from the third vaccination to enrollment was 147. Of the 607 individuals enrolled, 242 (40%) were vaccinated with a fourth dose, of which 74 (30%) became infected, and of the 365 (60%) that did not receive a fourth dose, 165 (45%) were infected (Table 1). We did not observe any cases of severe disease in this cohort. In the current analysis, we analyzed immune responses in blood samples collected at enrollment and day 30 and infections during the first 90 days of follow-up (Table 1). We analyzed outcomes at two-time points: 30 days post-enrollment and 90 days from the study start date. The median follow-up time was 76 days (IQR 75–77) for the four-dose group and 75 days (IQR 70–77) for the three-dose group. The baseline characteristics of the participants within each vaccination group are shown in Table 1.

### Vaccination with the fourth dose elicited binding and neutralizing antibodies against multiple SARS-CoV-2 variants

We measured the magnitude of IgA and IgG antibodies binding to multiple SARS-CoV-2 spike antigens using an antigen microarray at day 0 and day 30 of 212 individuals (Fig. 1a, b, Supplementary Data 1, Supplementary Fig. 1, Supplementary Data 2). Our arrays included spike antigens from the Alpha, Beta, Wuhan, Delta, Gamma, Iota,

**Table 1 | Demographic characteristics of the Clalit-Booster study participants**

|  | Overall | Three doses | Four doses | p |
|---|---|---|---|---|
| N | 607 | 365 | 242 |  |
| Sex, Male (%) | 170 (28.0) | 68 (18.7) | 102 (42.0) | <0.001 |
| Age (mean (SD)) | 47.25 (11.43) | 46.42 (10.95) | 48.49 (12.04) | 0.029 |
| Age group (%) |  |  |  | 0.002 |
| 18–34 | 86 (14.1) | 52 (14.2) | 34 (14.0) |  |
| 35–49 | 247 (40.6) | 156 (42.7) | 91 (37.4) |  |
| 50–64 | 235 (38.8) | 145 (39.7) | 90 (37.4) |  |
| 65+ | 39 (6.4) | 12 (3.3) | 27 (11.1) |  |
| Socioeconomic status (%) |  |  |  | 0.112 |
| Very High | 106 (17.5) | 52 (14.3) | 54 (22.2) |  |
| High | 243 (40.1) | 149 (41.0) | 94 (38.7) |  |
| Medium | 186 (30.9) | 118 (32.5) | 68 (28.4) |  |
| Low | 55 (9.1) | 33 (9.1) | 22 (9.1) |  |
| Very Low | 4 (0.7) | 4 (1.1) | 0 (0.0) |  |
| No data | 11 (1.8) | 7 (1.9) | 4 (1.6) |  |
| Occupation (%) |  |  |  | <0.001 |
| Physician | 155 (25.5) | 68 (18.6) | 87 (35.8) |  |
| Nurse | 159 (26.3) | 105 (28.8) | 54 (22.6) |  |
| Administration and support staff | 293 (48.2) | 192 (52.6) | 101 (41.6) |  |
| Medical Center (%) |  |  |  | <0.001 |
| Carmel | 86 (14.1) | 50 (13.7) | 36 (14.8) |  |
| Ha'emek | 126 (20.7) | 114 (31.2) | 12 (4.9) |  |
| Meir | 54 (8.9) | 40 (11.0) | 14 (5.8) |  |
| Soroka | 341 (56.2) | 161 (44.1) | 180 (74.5) |  |
| Days since third dose (median [IQR]) | 147.00 [140.00, 155.00] | 151.00 [139.00, 157.00] | 146.00 [142.50, 150.00] | 0.004 |
| PCR test count (median [IQR]) | 2.00 [1.00, 4.00] | 2.00 [1.00, 4.00] | 2.00 [0.50, 3.00] | 0.34 |

Kappa, Mu, Theta VOCs and multiple Wuhan spike antigens containing point mutations (Supplementary Data 5). We found that day 30 magnitudes of individuals receiving the fourth dose and were not infected by day 30 (n = 127) were significantly higher than baseline for the Wuhan strain for both IgG (p < 0.001) and IgA (p = 0.004), as well as for IgA to SARS-CoV-2 variants (p < 0.001). In contrast, IgG, but not IgA antibodies specific for the Wuhan strain (p < 0.001) waned in individuals that did not receive a fourth vaccination (n = 85), while both IgG and IgA reactive against the SARS-CoV-2 variants (p < 0.001), decreased in this group at the day 30-time point compared to enrollment. In addition, the decay of IgG binding to SARS-CoV-2 variants at day 30 was more pronounced than the decay of IgA responses (Fig. 1b).

To further assess the vaccine-elicited antibody responses, we selected a subset of 74 individuals for further in-depth immunogenicity assessment. Participants were selected based on their baseline immune history (BIH) to the Wuhan (vaccine) strain as measured upon enrollment. Specifically, we measured the IgG and IgA binding antibody levels to the RBD and S1 antigens of the Wuhan strain (Supplementary Data 3). We selected 38 individuals with the lowest BIH and 36 with the highest BIH antibody levels. Of these, 58 individuals received the fourth dose, and 23 were infected within the first 30 days (Supplementary Data 3-4). We found that uninfected fourth-dose individuals generated a significant rise in IgG and IgA titer against all four SARS-CoV-2 isolates measured by an ELISA against RBD of multiple variants (Fig. 1c p < 0.001, Supplementary Data 1). No significant rises

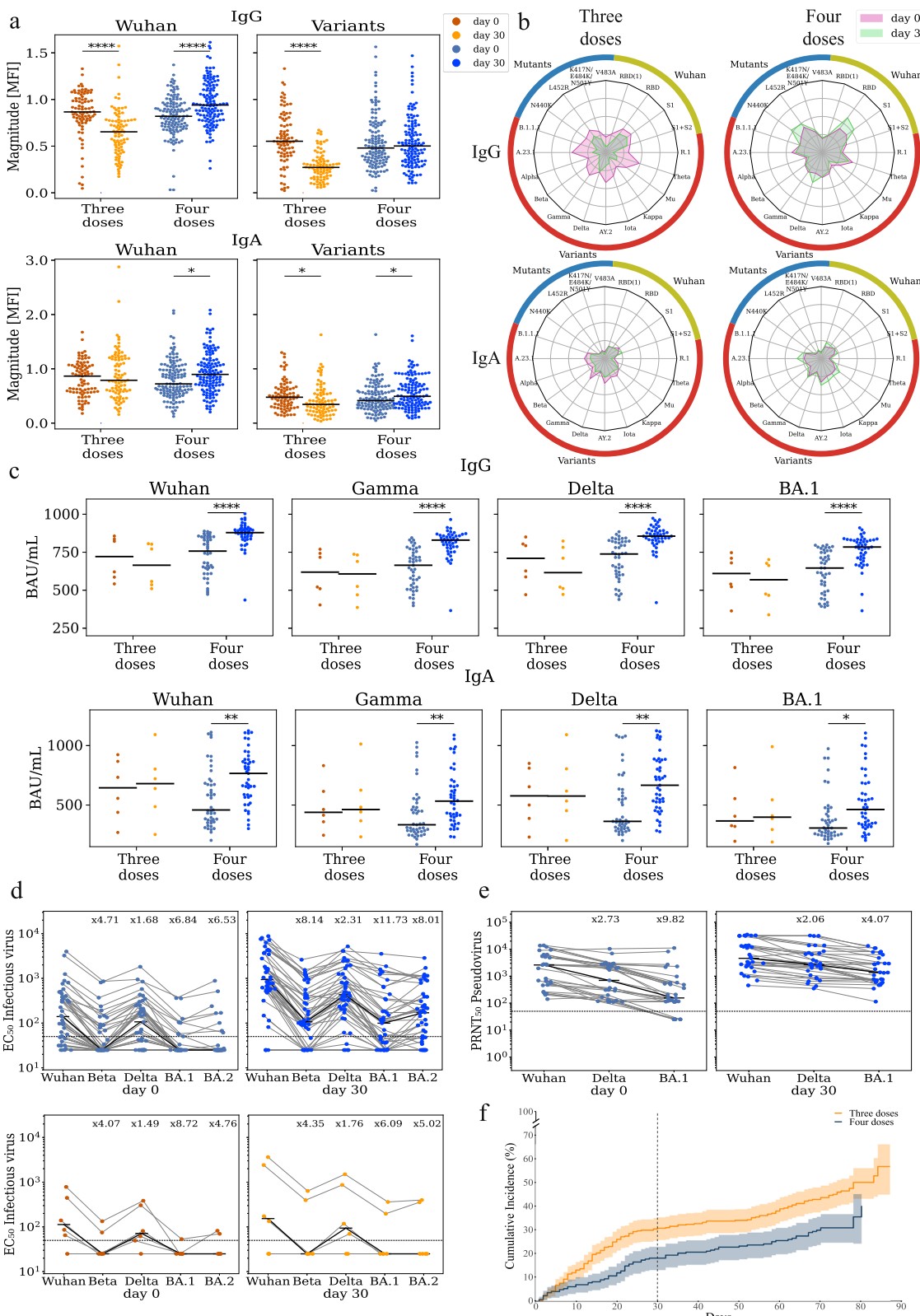

were observed in uninfected individuals who only received three doses.

We next characterized the neutralization titers of these 74 participants using infectious-virus neutralization assays against Wuhan, Beta, Delta BA.1, and BA.2 (Fig. 1d, Supplementary Fig. 2d and Supplementary Data 1). In line with previous studies, we found a significant reduction in EC50 neutralization titers to BA.1 and BA.2. At (baseline

fold drop of ×6.8 and ×6.5 respectively, post-vaccination fold drop of ×11.7 and ×8, respectively). The median baseline neutralization titer in both the three and four-dose groups was below the estimated protective threshold (EC < 50, Fig. 1d). We found a significant rise in the median neutralization titer of four-dose individuals at day 30 across all isolates (Fig. 1d $p < 0.001$). However, 17 participants (29%) failed to generate any measurable rise in neutralization titer to BA.1 on day 30.

**Fig. 1 | Vaccination with the 4th dose elicited binding and neutralizing antibodies against SARS-CoV-2.** Responses of uninfected participants were analyzed at enrollment (day 0) and at day 30 using multiple serological assays. **a** IgG and IgA magnitude to antigens from the Wuhan strain and SARS-COV-2 variants of concern including Alpha, Beta, Gamma, Delta, Iota, Kappa, Mu, Theta, and several sub-variants (see Supplementary Data 5). Antigen microarrays spotted with receptor binding domain (RBD), S1 and spike proteins of the Wuhan vaccine strain and multiple other variants of concern were used to measure the magnitude of responses at day 0 (enrollment) and day 30 post enrollment $n = 212$ biologically independent samples. Black lines denote the median. **b** Spider plots depicting the enrollment (pink) and day 30 (green) antibody levels to Wuhan antigens (gold), variants of concern (red) and RBD mutants (blue). The average normalized magnitude to each antigen is plotted in individuals that received 3 or 4 doses. **c** IgG and IgA anti RBD ELISA binding titers for a subset of 51 uninfected participants. Black lines denote the median. **d** Infectious virus neutralization half maximal effective concentration (EC50) titers of the same individuals in **c**. top - uninfected individuals that received 4 doses $n = 44$ (blue). bottom - uninfected individuals that received 3 doses $n = 6$ (orange). Black lines denote the median. **e** Pseudovirus neutralization titers of uninfected individuals that received 4 doses $n = 30$ biologically independent samples (blue). Black lines denote the median. **f** Cumulative incidence of SARS-CoV-2 infections in participants receiving three doses $n = 365$ vs. four doses $n = 243$ biologically independent samples of the Pfizer vaccine. Four doses of the vaccine significantly reduced infection rates at day +30 (HR = 0.55, $p = 0.002$) and across all interim followup time (HR = 0.63, $p = 0.003$) as compared to three doses. The line represents cumulative incidence, and shaded bands denotes the 95% confidence intervals. *P*-values were computed using the two-sided wilcoxon rank-sum test. $*p < 0.05$; $**p < 0.001$; $****p < 0.00001$.

We also characterized 47 participants from the immunogenicity subset using pseudovirus neutralization assays against the Wuhan, Delta, and BA.1 strains. We found a significant rise in titers at day 30 following the fourth dose across all three isolates (Fig. 1e, Supplementary Fig. 2e and Supplementary Data 1). In contrast to the infectious-virus neutralization assay, all of the 30 four-dose and uninfected participants had measurable neutralization titers against BA.1 at day 30.

We then compared the IgG and IgA antibody responses of breakthrough infections in individuals that were infected within the first 30 days from enrollment (Supplementary Fig. 2). We found that overall infected participants had a significant rise in antibody levels regardless of whether they received a fourth dose of the vaccine or not. Moreover, there were no significant differences between day 30 titers of infected participants with 3 and 4 doses of the vaccine (Supplementary Fig. 2).

### A fourth dose of the Pfizer-BioNTech mRNA vaccine significantly reduced the risk of symptomatic SARS-CoV-2 infection

To assess the effect of the fourth dose on symptomatic SARS-CoV-2 infection, we estimated vaccine efficacy (VE) using a Cox model adjusted for age, occupation, medical center, and time from the third vaccination. We found that the VE at day 30 was 45.5% [95% CI, 19–63%], and the VE at the interim time point was 37% [95% CI, 15–53%] (Supplementary Table 1-2). Similar estimates were also obtained from a Poisson regression model adjusted for the proportion of daily positive PCR tests in Israel (Supplementary Table 3-4). Cumulative incidence curves of SARS-CoV-2 infection in the fourth dose group vs. the three-dose group are shown in (Fig. 1f)

### Baseline binding antibody markers are associated with neutralizing antibody titers

Upon enrollment, we profiled the IgG and IgA baseline immune history to multiple SARS-CoV-2 antigens (Supplementary Data 1, Supplementary Data 5) in all trial participants ($n = 607$). These profiles were used to rank individuals into three BIH groups: low, mid, and high (Fig. 2a). We compared the breadth and magnitude of baseline and day 30 responses of IgG and IgA low-BIH and high-BIH individuals that received a 4th dose and found that individuals in the low-BIH group generated weaker and narrower antibody profiles post-vaccination as compared to the high-BIH group (Fig. 2b). We found that baseline IgG and IgA antibody profiles of individuals within the low and high-baseline groups were significantly different from one another (Fig. 2b). To quantify this further, we computed the correlations between IgG and IgA magnitudes at baseline. We found that magnitudes were only moderately correlated to one another ($r < 0.310$, Supplementary Fig. 3).

We hypothesized that individuals from the low-baseline group would have significantly lower baseline neutralization titers. We found significant differences in the neutralization titer of the low-baseline and high-baseline groups at day 0 for the Wuhan, Beta, Delta, and Omicron BA.1 strain (Fig. 2c). Notably, 20% of the individuals in the low group had no detectable neutralization titers to the Wuhan strain. None of them had any detectable titers to the Omicron BA.1 strain (Fig. 2c). In the high-baseline group, 14 individuals had no detectable titers to the Omicron BA.1 strain (Fig. 2c). However, post-vaccination with the fourth dose, neutralization titers were not significantly higher in the high-baseline group than in the low-baseline group (Fig. 2c). In line with our overall observation of significant waning of individuals in the three-dose group, we found out that ranking individuals who were un-infected by day 30 by both IgG or IgA responses to VOCs, IgG antibody responses waned more significantly in the high-baseline group vs. the low-baseline group (IgG $p < 0.001$; IgA $p < 0.001$, (Fig. 2d). Interestingly, IgA responses waned significantly less than IgG responses, especially in the high baseline groups (Fig. 2d).

### Baseline binding IgA and IgG responses are correlates of protection for the Pfizer-BioNTech mRNA vaccine

We hypothesized that individuals with a low-baseline immune history to SARS-CoV-2 might be at an increased risk for SARS-CoV-2 infection. We found that in the three and four-dose groups, the baseline IgA responses against the Wuhan RBD were significantly higher in uninfected individuals as compared to infected individuals ($p = 0.042$ and $p = 0.042$, Fig. 3a). IgG responses against the RBD were not significantly associated with infection status (four doses: $p = 0.083$; three doses $p = 0.281$). IgA responses to the S1 protein were significantly higher in uninfected individuals as compared to infected individuals in the three-dose group ($p = 0.032$, Fig. 3a), and responses to VOCs were higher in uninfected individuals in the four-dose group ($p = 0.048$ Fig. 3a).

To analyze baseline serological markers as COPs our primary analysis focused on the following antibody binding measures: (1) Wuhan magnitude−average response to the RBD, S1, and full-length S protein of the Wuhan strain; (2) Variants magnitude−average response to multiple SARS-CoV-2 variants of concern including Alpha, Beta, Wuhan, Delta, Gamma, Iota, Kappa, Mu and Theta; (3) RBD mutants magnitude−average response to multiple RBD mutants (Supplementary Data 5); (4) Wuhan S2 IgG levels (RAD Bioplex); (5) Wuhan RBD IgG levels (Abbott Alinity). Magnitude baseline markers were computed for both IgG and IgA separately. Individuals were ranked and divided into three groups using quartiles as described above (Fig. 2a).

We compared the infection rates of the three and four-dose groups at two time points: 30 days post-vaccination and at the interim analysis time point, which included 60−90 days of followup for all participants. We used a Cox regression model to compute the hazard ratios by comparing the low-baseline to the high-baseline groups. At the 30-day time point, IgA magnitude to the Wuhan strain was associated with infection risk in fourth dose recipients (HR = 3.19, $p = 0.019$, Fig. 3b, Fig. 4a, Supplementary Table 5). IgA magnitude to SARS-CoV-2 VOCs was more strongly associated with infection risk (HR = 4.45,

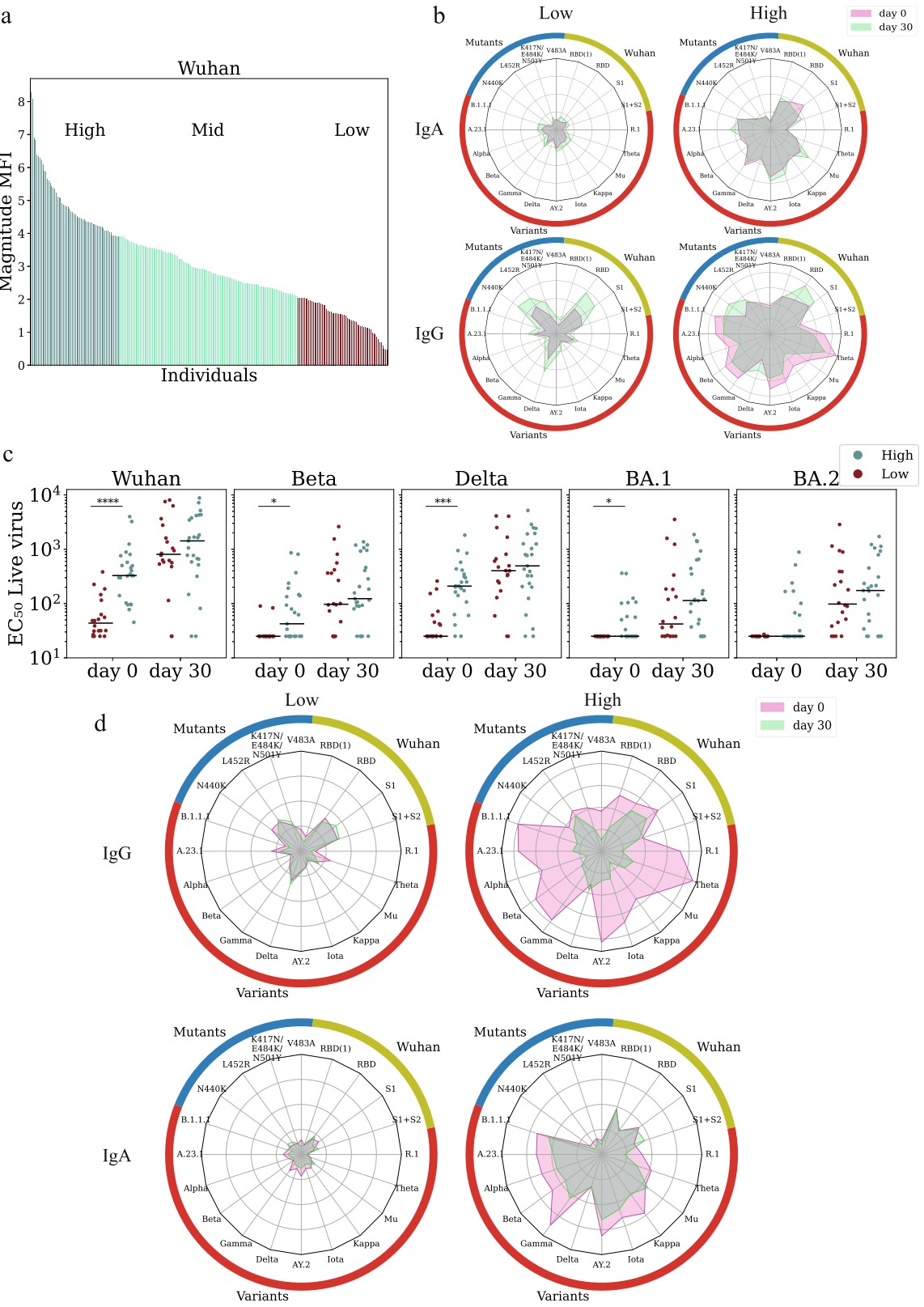

*p* = 0.006, Figs. 3b and 4b). None of the IgG baseline markers were significantly associated with infection risk in the fourth dose vaccine group. However, in the three dose group, IgG levels as measured using the Abbott Alinity assay were associated with infection risk (HR = 1.59 *p* = 0.02), and a similar trend was observed for the Rad Bioplex S2 assay (HR = 1.39, *p* = 0.089; Figs. 3b and 4c). We then evaluated infection risk at the 60–90 day time point (Fig. 4d, Supplementary Table 6). At the

60–90 day follow-up time point, we found that in fourth dose recipients, all markers were associated with infection risk (IgA Wuhan: HR = 2.05, *p* = 0.041; S2: HR = 1.77, *p* = 0.025; Alinity: HR = 1.65 *p* = 0.049, Fig. 4). Within the third dose group the Bioplex S2 and Alinity were significantly associated with infection risk (S2: HR = 1.45, *p* = 0.022; Alinity: 1.54, *p* = 0.008; Fig. 4). In our secondary analysis, we considered additional individual antigens as baseline correlates, and

**Fig. 2 | Ranking individuals using baseline binding antibody markers is associated with baseline neutralizing titers. a** Ranking of 242 vaccinated individuals by their magnitude to SARS-CoV-2 Wuhan at enrollment. Each bar represents the magnitude of a single participant defined as the average response to the set Wuhan antigens (see Supplementary Data 5). Participants were divided into low (lowest quartile); mid (quartiles 2 + 3); and high (highest quartile) based on magnitude of IgA responses to Wuhan. **b** Spider plots of the average normalized responses in the low-baseline immune history (BIH) and high-BIH groups to a set of spike and receptor binding domain (RBD) proteins including the Wuhan spike and RBD, RBD mutants, and multiple SARS-CoV-2 variants of concern (see Supplementary Data 5)

spike proteins. Responses of the low and high response groups of 127 uninfected individuals that received a fourth boost are plotted separately for IgA (top) and IgG (bottom). **c** Infectious-virus neutralization titers of 45 vaccinated uninfected individuals at day 0 and day 30 from the low-baseline (red) and high-baseline (teal) groups. Black lines denote the median. *P*-values were computed using the two-sided wilcoxon ranksum test.*$p < 0.05$; ***$p < 0.0001$; ****$p < 0.00001$. **d** Average IgA and IgG spider plots of 85 individuals that received 3 doses of the vaccine at day 0 (pink) and day 30 (green). Individuals were sorted by baseline response to SARS-CoV-2 variants of concern (VOC).

identified several additional baseline markers associated with infection status in both groups (Supplementary Data 6-7).

## Combinations of IgG and IgA baseline markers as correlates of protection

Given the moderate correlations between IgG and IgA magnitudes (Supplementary Fig. 3), we reasoned that combinations of IgG and IgA markers might provide improved COP against symptomatic SARS-CoV-2 infection. For each pairwise combination of baseline markers, we intersected the low-baseline and high-baseline groups and compared the infection rates of these groups. We found that combinations of baseline markers were more strongly associated with infection risk in both the three dose and four dose groups than single baseline markers (Fig. 5, Supplementary Table 7-8). For example, at the 60-90 day followup time point the infection rate in the low-baseline group ranked by IgG levels to RBD mutants, and IgA levels to VOCs, was 42.1% and only 13% in the high-baseline group in the fourth dose group (HR = 8.18, $p = 0.018$). The same marker was also associated within the third dose group (HR = 6.34, $p = 0.008$) (Fig. 5a, b). Multiple marker combinations were significantly associated with infection status in both groups (Fig. 5c). The combination of baseline IgA responses to the Wuhan strain and IgA responses to VOCs were significantly associated with infection risk at both timepoints, however, the association was stronger at the day 30 timepoint in fourth dose recipients (day 30 HR = 5.73, $p = 0.009$, interim time point: HR = 2.34, $p = 0.051$, (Fig. 5c). The combination of IgG Alinity RBD and IgG Bioplex S2 assays was also associated with infection status at the two time points for the third dose group and interim time point for the fourth dose group (Fig. 5c). Similar estimates for single markers and their pairwise combinations were obtained using a Poisson regression model as previously described (Supplementary Table 9-12).

## Validation of baseline IgG and IgA binding as correlates of protection

To further validate the use of baseline binding IgG and IgA markers as correlates of protection for SARS-CoV-2 infection, we used the same framework presented above for analyzing an independent clinical cohort that followed 46 healthy adults over 9 months between October 2021 and July 2022 (Supplementary Table 13). Baseline and monthly blood samples were collected from all study participants, and nasopharyngeal swabs were collected every week. Participants also reported symptomatic sickness events, including SARS-CoV-2 infections (see methods for details). During this period, 72% ($n = 33$) of the cohort was infected with SARS-CoV-2. Participants had diverse SARS-CoV-2 immune histories, which included Pfizer BNT162b2 vaccination (2–3 doses). A single participant who had a previous SARS-CoV-2 infection was removed from our analysis. We used baseline plasma samples to rank individuals based on the same IgG and IgA markers used above (see methods for details). We then compared the infection rates in the low- and high-baseline groups for each marker and all combinations of IgG and IgA markers (Supplementary Data 8). The analysis was conducted at two time points: (1) in April 2022 after the BA.1 omicron wave in Israel (Supplementary Fig. 4, Supplementary Table 14); and (2) in June 2022 after a second BA.4/5 omicron wave in

Israel (Fig. 6, Supplementary Table 15). A Cox regression model was used to estimate hazard ratios. We identified both IgG and IgA baseline markers that were associated with protection: Ranking individuals using baseline IgG and IgA magnitude to Wuhan antigens was significantly associated with protection (HR = 4.18, $p = 0.007$; HR = 2.84, $p = 0.036$, respectively, Fig. 6b). Baseline IgG ranking using magnitude to SARS-CoV-2 VOCs was associated with protection (HR = 3.74, $p = 0.006$, Fig. 6b). Still, baseline IgA was not significantly associated ($p = 0.368$). Furthermore, a combined ranking by IgG magnitude to VOCs and IgA magnitude to Wuhan was most significantly associated with protection (HR = 8.62, $p = 0.002$).

## Discussion

The main goal of this study was to identify novel binding antibody correlates of protection against symptomatic SARS-CoV-2 infection during the omicron wave in vaccinated healthy individuals following one or two booster doses. Following the fourth dose, we showed that antibody levels were significantly elevated, which correlated well with neutralization titer against multiple SARS CoV-2 variants and overall infection protection. We found that IgA baseline markers against RBD mutants and spike VOCs are strongly associated with protection.

In our primary analysis, we used a set of five baselines IgA and IgG binding antibody markers, two of which are commercially available IgG assays. Using a simple quartile approach to define the low- and high-baseline response groups, we showed that infection rates in the low-response group were significantly higher than in the high-response group for both third and fourth dose recipients. Due to the moderate correlation between IgG and IgA baseline antibody levels of these five markers, we next considered pairwise combinations of these five baseline markers, including an IgG and IgA marker. By ranking individuals using such pairs of markers, the differences in infection rates between the low-baseline and high-baseline groups were more pronounced than single markers. This suggests that both IgA and IgG antibodies may play a protective role in preventing SARS-CoV-2 symptomatic infection. We also found that the combinations of IgA to Wuhan and variants were associated with infection risk in the fourth dose group at both time points and that all different combinations of IgG pairs were associated with infection status at the interim followup time point. Of particular interest is the combination of the Alinity RBD IgG and Rad Bioplex S2 IgG assays, both clinically approved assays widely used in clinical virology labs, and were COPs in both groups.

To validate our findings, we used a second independent clinical cohort. While infection rates within this cohort were high (72%), identifying statistically significant COPs in such a small cohort ($n = 46$) is highly challenging. We found that IgA and IgG baseline binding antibody markers and their combinations were correlates of protection for SARS-CoV-2 infection. Using the best IgG and IgA combination marker all (100%) of the individuals in the low-baseline group were subsequently infected, as compared to 55% of the high-BIH group. These data highlight and strengthen the generalizability of our findings, and suggest that ranking individuals by baseline immune-history using binding antibody profiles may be a viable alternative to using neutralization assays and other functional assays as correlates of protection.

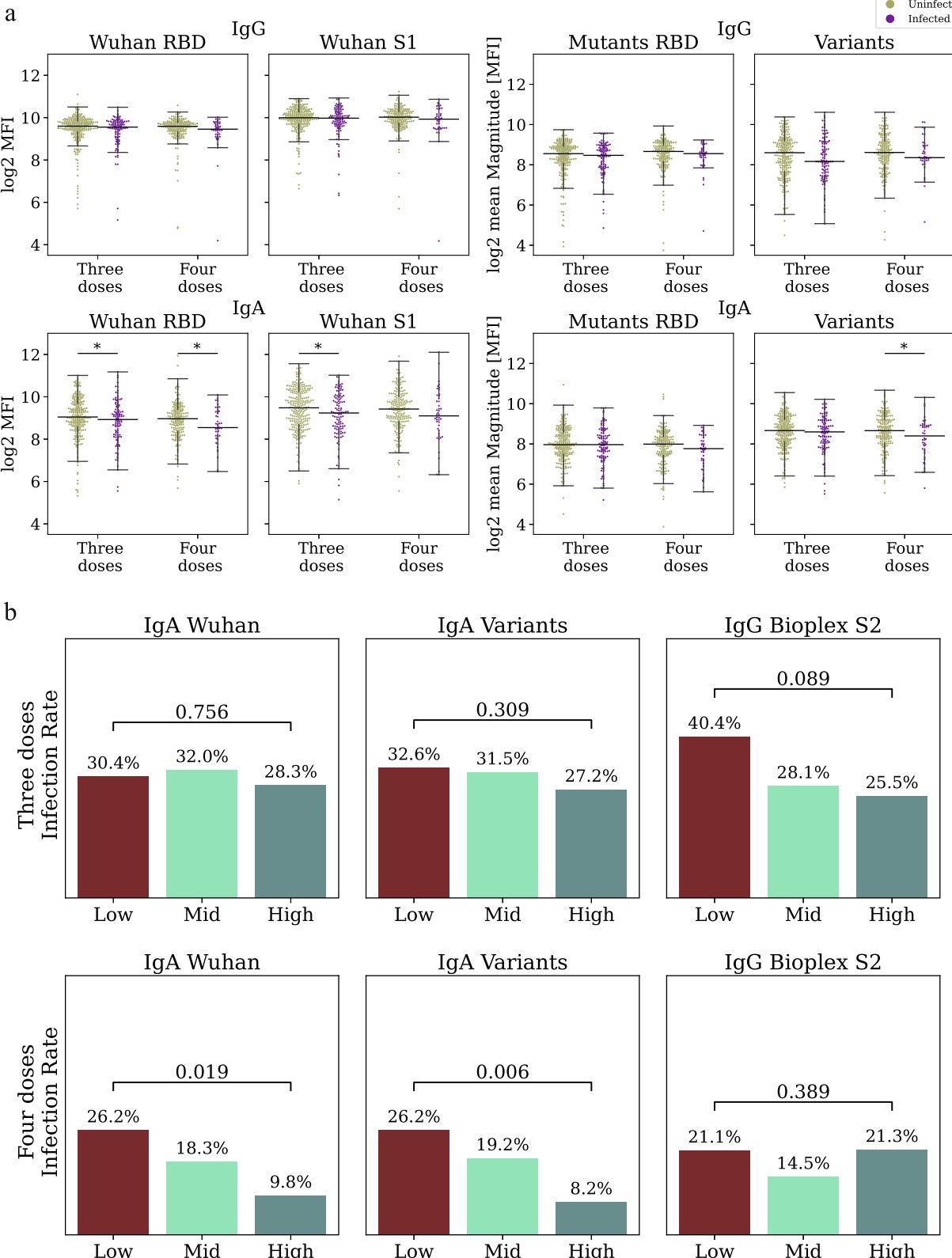

**Fig. 3 | Identifying baseline correlates of protection following three or four doses of the Pfizer-BioNTech vaccine. a** Antibody levels of uninfected (*n* = 451 brown) and infected (*n* = 156 purple) individuals against Wuhan, receptor binding domain (RBD) mutants and variants of concern (VOC) measured at enrollment. Black lines represent the median, and whiskers represent 1.5 times the interquartile range. P-values were computed using the two-sided wilcoxon ranksum test.*$p$ < 0.05. **b** Day 30 infection rates in low- mid- and high-baseline response groups ranked by baseline binding antibodies. Comparisons were conducted for three dose (top) and four dose (bottom) recipients separately. Individuals were ranked by IgA magnitude to Wuhan (Left), IgA to SARS-CoV-2 variants (Center, Supplementary Data 5), and by IgG S2 Bioplex (Right) *P*-values were computed using a cox proportional hazard model, adjusted for age, occupation, medical center, and time from the third vaccination.

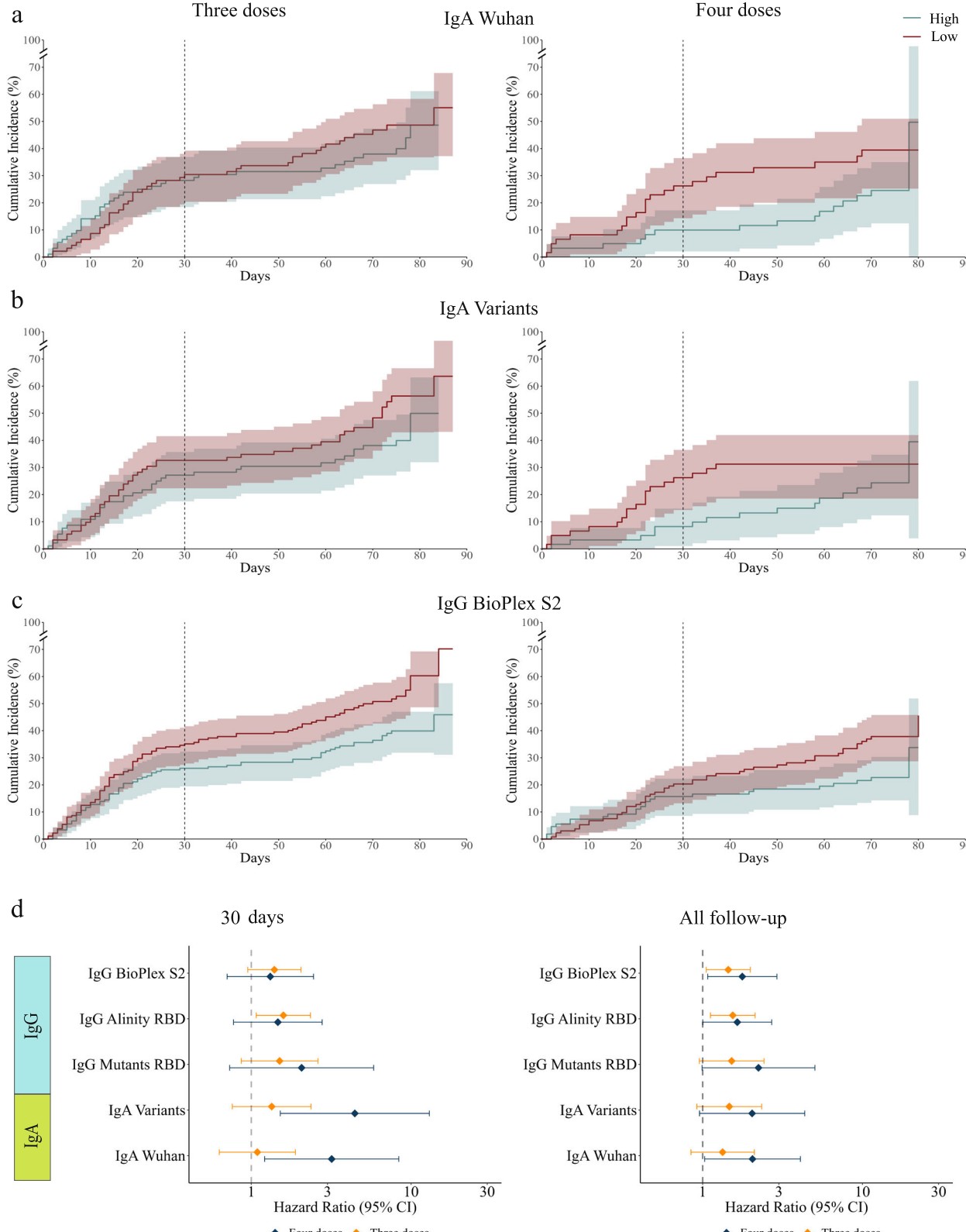

**Fig. 4 | Cumulative incidence plots of individuals in the low and high-baseline response groups as measured using. a** IgA response to Wuhan. The line represents cumulative incidence, and shaded bands denotes the 95% confidence intervals. **b** IgA response to SARS-CoV-2 variants of concern (VOC). **c** IgG response to the S2 protein (RAD bioplex assay). **d** Hazard ratios for the five primary baseline markers comparing low to high baseline response groups for individuals vaccinated with three doses $n = 365$ for IgG BioPlex S2 and IgG Alinity receptor binding domain (RBD), $n = 184$ for IgG Mutant RBD, IgA Variants and IgA Wuhan (orange) or four doses $n = 242$ for IgG BioPlex S2 and IgG Alinity RBD, $n = 122$ for IgG Mutanta RBD, IgA Variants and IgA Wuhan (blue) at day 30 (left) and the interim followup time point (right). The dot represents the hazard ratios, error bars denote the 95% confidence intervals. Hazard ratios were computed using a cox proportional hazard model adjusted for age, occupation, medical center, and time from the third vaccination.

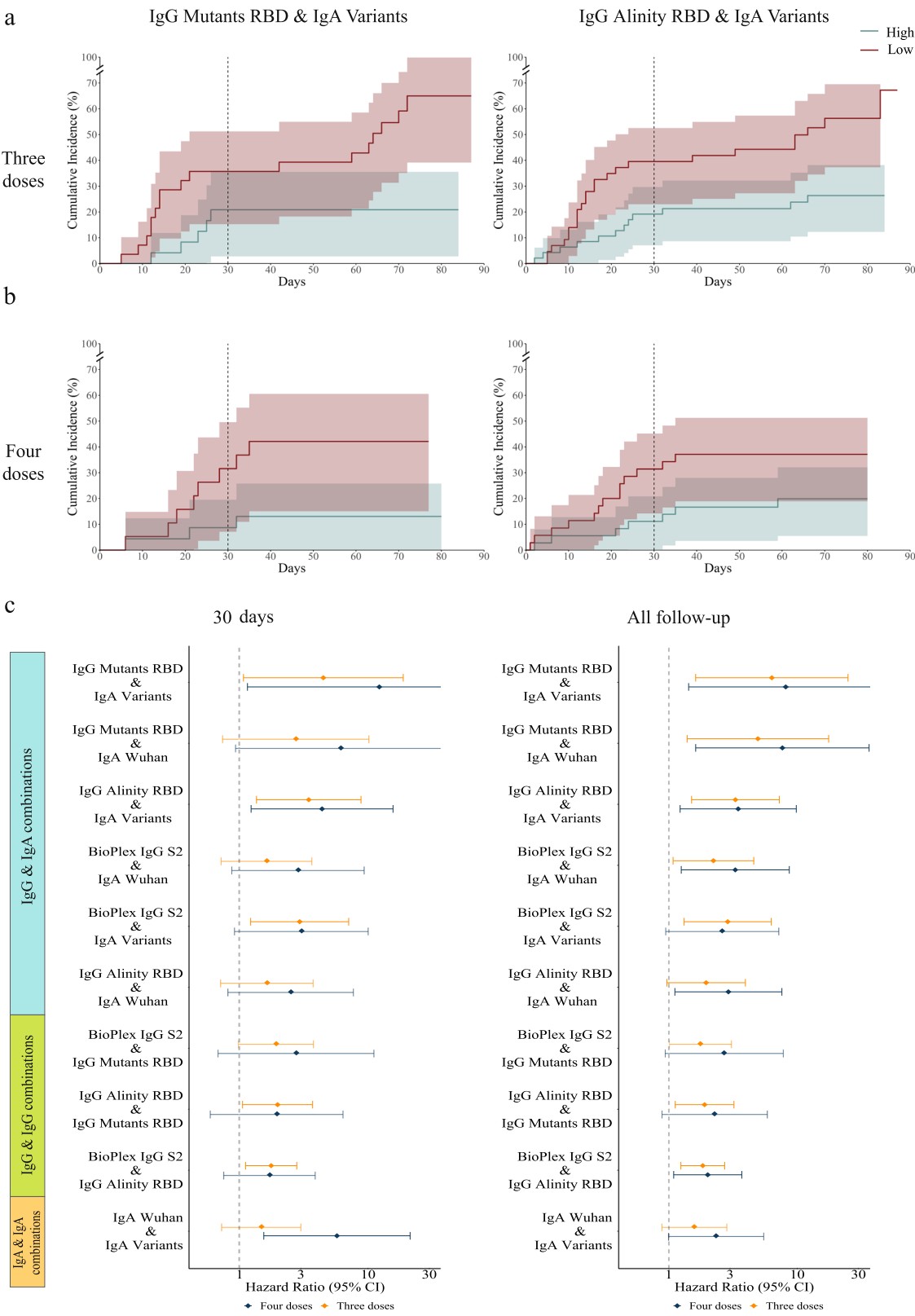

Studies on COPs for the Pfizer-BioNTech vaccine found that neutralizing antibody titers were a correlate of protection following two doses of the Pfizer-BioNTech vaccine[26–29] and this was observed as well for other vaccines[16,30–32]. To the best of our knowledge, this is the first study to report COPs for booster doses of the Pfizer vaccine, and the first study to report COPs against infection with Omicron B.A1[27]. The COPs reported here were based on baseline binding antibody levels and not on neutralizing antibody titers. Binding antibodies measured by ELISA IgG antibody titers have been previously shown to be COPs for influenza[33] and for SARS-CoV-2[34]. A clear advantage of binding antibody assays is that they can readily be measured at baseline for large cohorts, unlike neutralizing antibody assays. Other recent studies have highlighted the role of baseline immunological and host features on the response to vaccines[35–37].

**Fig. 5 | Combinations of IgG and IgA baseline markers are improved baseline correlates of protection.** Pairs of baseline markers were used for ranking individuals using the intersection between the low and high groups of each baseline marker separately. Comparisons were conducted for 3rd and 4th dose recipients separately. **a, b** Cumulative incidence plots of individuals in the low and high-baseline response groups as measured using: **a** IgG receptor binding domain (RBD) mutants and IgA SARS-CoV-2 variants of concern (VOC) (four doses n = 42, three doses n = 52); and **b** IgG Alinity and IgA SARS-CoV-2 VOCs (four doses n = 71, three doses n = 90). The line represents cumulative incidence, and shaded bands denotes

the 95% confidence intervals. **c** Hazard ratios comparing low to high baseline response groups using pairwise combinations of baseline binding antibody markers for individuals vaccinated with three doses (orange) or four doses (blue). The dot represents the hazard ratios, error bars denote the 95% confidence intervals. Hazard ratios were computed using a cox proportional hazard model adjusted for age, occupation, medical center, and time from the third vaccination. A number of individuals per vaccine group is different in each pairwise combination markers and is indicated in Supplementary Table 16.

IgA-based COPs have not been studied extensively, mainly due to the lack of standardized assays to quantify IgA levels in mucosal samples and their relatively low concentration in the blood. A recent study in healthcare workers reported that serum IgA levels were associated with protection from symptomatic infection[38]. Burt et al.[39] found that both serum IgG & Mucosal IgA are important COPs against symptomatic influenza infection in a human challenge trial. They further showed that combinations of HAI titers and mucosal IgA titers were improved COPs against influenza infection. While IgA antibodies that offer protection from SARS-CoV-2 infection are primarily found in the mucosa[40], our reported IgA correlates were measured from the serum. A recent study in celiac patients reported that the gut mucosal and serological IgA repertoires share strong clonal overlap despite originating from different plasma cell compartments[41]. Similar findings were reported for IgA plasmablasts from the serum and lungs following influenza vaccination[42]. These studies suggest that while the serum IgA repertoire may not be directly involved in protection from infection of the respiratory tract, it may correctly reflect the mucosal IgA repertoire.

Our study measured the levels of a wide variety of antibodies against VOCs and against RBD mutants as COPs. Previous studies of binding antibody COPs were based on ELISA titers to a single viral variant, requiring one to choose a relevant variant. The approach used here utilizes a cross-reactivity score that integrates across all previous SARS-CoV-2 VOCs (excluding Omicron B.A1 or B.A2). The IgA and IgG magnitude to SARS-CoV-2 VOCs and to RBD mutants are measurements of the cross-reactive binding antibody responses. Due to the rapid evolution of SARS-CoV-2, it is difficult to identify the optimal single marker or variant which may best predict protection from infection for a novel VOC. Our data suggest that by using aggregate cross-reactivity measures to multiple variants, we can obtain more robust COPs even for strains that are antigenically distinct from previous strains, such as the Omicron B.A1 strain[7].

We show that ranking individuals using IgA & IgG markers was associated with significant differences in neutralizing antibody titers. Individuals with low-baseline binding antibodies had significantly lower neutralizing antibody titers than the individuals in the high-baseline group. Interestingly, while 35% of the individuals in the low-baseline group failed to develop detectable neutralizing antibody titers following a fourth dose, the majority of participants (65%) generated a significant rise in neutralizing antibody titers at day 30, demonstrating their ability to mount an adequate immune response following a fourth booster dose. In fact, on day 30, we found no significant differences between the individuals in the low-baseline and high-baseline groups, indicating that the fourth dose induced steeper rises in neutralizing antibody titers in the low-baseline group. These data suggest that most individuals in the low-baseline group may have low-baseline titers not due to a lack of ability to respond but possibly due to increased decay rates of their circulating anti-SARS-CoV-2 antibodies. Despite the significant rise in neutralizing antibody levels following the fourth dose, the number of infections in the low-baseline group (n = 16) was significantly higher than in the high-baseline group (n = 7, 43% vs. 20%, p = 0.051).

We also demonstrated that a fourth dose of the vaccine generated a significant rise in both IgG and IgA binding antibodies, and

neutralizing antibody titers using both infectious-virus and pseudo-virus assays. At baseline, 59 (79.73%) of the 74 individuals from the immunogenicity subset had low detectable neutralizing titers to the Omicron B.A1 strain, and 56 (75.67%) had no titers to the Omicron B.A2 strain. Thirty days post-vaccination, only 13 (29%) of the 45 vaccinated uninfected individuals within the immunogenicity subset had no detectable titers to the B.A1 variant. These results are consistent with data reported by Regev-Yochai et al.[13] who reported a significant rise in neutralization titers 14 days post-vaccination with the fourth dose to both Delta and Omicron B.A1 variants. This rise in binding and neutralizing antibody titers was also associated with increased protection against symptomatic Omicron B.A1 infection, but this effect was transient. This suggests that the additional boosters should be administered at the onset of new infection waves where they may be important for reducing the spread of a new variant. It is also important to note that individuals who did not receive a fourth dose were a median of 177 days from their third dose. Therefore, it is possible that the protective effect of the 4th dose may be due to the shorter time interval from their last vaccination and not due to the number of vaccine doses received.

Our study also found that the antibody responses of individuals who only received three doses of the Pfizer-BioNTech vaccine continued to significantly wane over the first 30 days of the trial. This data is in agreement with a previous study that reported declines in binding and neutralizing titers up to 6 months post the second dose of the Pfizer-BioNTech vaccine[43]. However, we found that IgG antibodies waned more significantly than IgA antibodies. We also found that individuals with hybrid immunity - i.e. that received 3 or 4 doses of the Pfizer-BioNTech vaccine and were subsequently infected with B.A1 had significant rises in neutralization titers to all five VOCs. A recent study reported that individuals with hybrid immunity had increased protection from Omicron infection as compared to vaccination alone[44].

Our study identified COPs for symptomatic SARS-CoV-2 infection. However, there is significant evidence of frequent asymptomatic SARS-CoV-2 infections. To assess the extent of such infections in our cohort, we analyzed anti SARS-CoV-2 nucleocapsid antibody levels as measured using the Rad-BioPlex assay (Supplementary Fig. 5). We found that 2.9% of the uninfected individuals had detectable NC antibody levels at day 30, suggesting that indeed there were additional asymptomatic infections in our cohort. However, many recent studies have outlined that not all natural infections induce anti-NC antibodies[45,46] and that they also wane quite rapidly in some individuals[47–49], suggesting that not all asymptomatic infections can be detected using anti-NC antibodies.

One of the limitations of our study is that these findings are based on antibody magnitudes measured using an antigen array-based assay, which are currently not widely used in clinical settings. However, multiple other studies used this assay to profile SARS-CoV-2 antibody responses[50–57], and these arrays are now commercially available. Furthermore, we also found that the combination of IgG levels measured using the Alinity RBD assay and S2 levels measured using the Biorad Bioplex assay was a correlate of protection in both the third and fourth dose groups. While the association with infection risk was weaker for this combination as compared to combinations of IgG and IgA markers, these assays can be readily used by clinical labs to identify

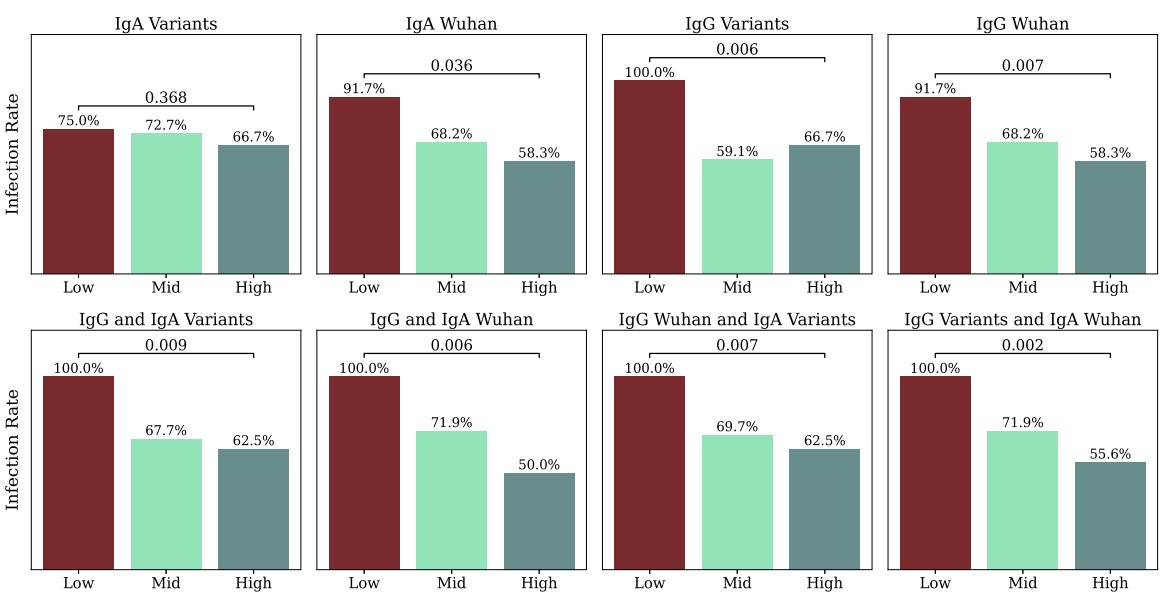

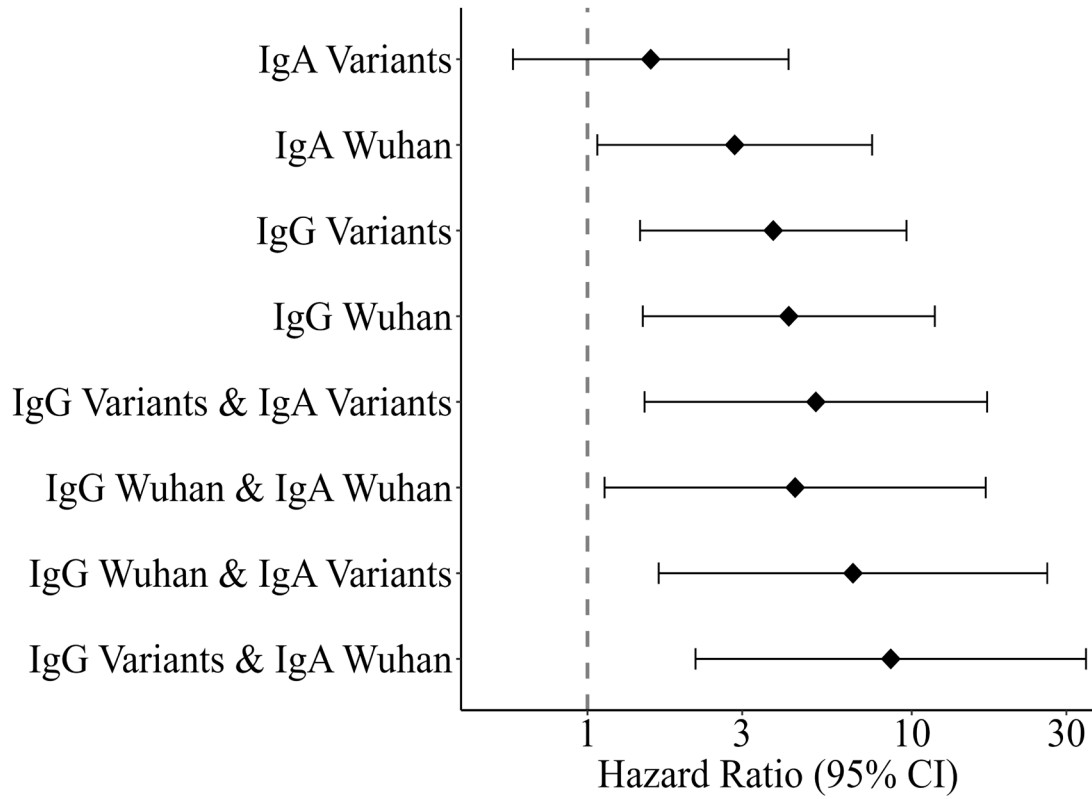

**Fig. 6 | Baseline correlates of protection in a validation cohort.** An independent cohort of 46 individuals was followed for 290 days. Individuals were ranked by several baseline binding antibody markers into low- mid- and high response groups, and SARS-CoV-2 infection rates of each group were compared. **a** Infection rates in the low- mid- and high-baseline response groups based on: IgA magnitude to SARS-CoV-2 variants including Alpha, Beta, Gamma, Delta, Iota, Kappa, Mu, Theta, and several sub-variants (see Supplementary Data 5), IgA magnitude to Wuhan, IgG magnitude to SARS-CoV-2 variants, IgG magnitude to Wuhan (top row) and their combinations (bottom row). *P*-values were computed using a cox proportional hazard model, adjusted for for age, sex and number of vaccine doses. **b** Hazard ratios for the four primary baseline markers ($n = 24$) and their combinations comparing low to high baseline response groups ($n = 13$–15). The dot represents the hazard ratios, error bars denote the 95% confidence intervals. Hazard ratios were computed using a cox proportional hazard model adjusted for age, sex and number of vaccine doses.

individuals with low baseline immune history against SARS-CoV-2 that are at an increased risk of infection.

In conclusion, our study demonstrated that combinations of IgA and IgG baseline antibody levels to SARS-CoV-2 VOCs are associated with protection from symptomatic infection. Importantly, our study identified a subpopulation of healthy adult individuals with low-baseline levels of IgA and IgG who are at increased risk for SARS-CoV-2 infection, despite receiving three or four doses of the Pfizer-BioNTech vaccine. Additional studies are required to assess whether this subpopulation is also at an increased risk for severe disease, and whether it may spread infection more readily than others. While the underlying mechanism for the increased susceptibility to symptomatic infection in this subpopulation is currently unknown, our study found that these individuals are indeed capable of mounting neutralizing antibody titers following an additional booster shot, suggesting that other functional differences between these groups such as Fc effector functions and antibody waning dynamics may be at play. These findings warrant further longitudinal functional studies of this group across longer followup time.

## Methods

### Study design and setting

This Clalit HCP Booster study is an ongoing prospective cohort study designed to assess the association between different serological profiles and risk for SARS-CoV-2 infection, comparing those vaccinated with three doses of Pfizer-BioNTech vaccine (Three-dose) to those who received a fourth booster dose (Four-dose). For this multicenter study, we enrolled HCPs at four medical centers managed by Clalit Health Services (CHS), the largest integrated payer-provider healthcare organization in Israel with 4.7 million members. The medical centers are spread across Israel: Ha'Emek and Carmel Medical Centers in northern Israel, Meir Medical Center in the central region, and Soroka University Medical Center in southern Israel.

We enrolled 639 HCPs over 18 years of age. All participants received a primary vaccine series of two doses and a third dose six months later. The third dose was given at least three months prior to enrollment. We excluded 19 individuals with prior SARS-CoV-2 infection or a history of receiving chemotherapy or immunosuppression therapy within the last three months. Nine participants had missing data, and four participants dropped out of the study.

Data related to all SARS-CoV-2 PCR tests in Israel is collected centrally by the Israeli Ministry of Health (MoH) and is updated daily into CHS's electronic medical records. We, therefore, collected data directly from the CHS database. Additionally, all participants completed a brief questionnaire at enrollment and at every monthly visit. Data collection and management for the study were conducted using REDCap (Vanderbilt University, Nashville, TN, USA).

Followup time was calculated in person-days. At the interim time point analyzed here, participants were followed for three months. The time to infection for individuals receiving three doses was measured from the day of enrollment and for individuals receiving four doses of vaccine, starting from the eighth day after receiving the fourth vaccine dose.

### Selecting low and high baseline immune history groups

Our final cohort included 607 healthcare providers older than 18 years from 4 medical centers in Israel. A subset of 74 participants with lowest and highest antibodies level selected for further in-depth immunogenicity assessment. 341 individuals of the main cohort were analyzed in preliminary analysis. IgG and IgA antibodies binding level has been tested against the S1 and RBD proteins of the Wuhan strain.

### Validation cohort

To assess the reproducibility and generalizability of our results we utilized the same baseline binding antibody markers to analyze an independent validation cohort. Specifically, we used baseline samples from a separate longitudinal study that followed 50 individuals across 9 months, in which baseline and monthly blood samples were collected. In addition, detailed infection history was monitored using weekly nasal swabs. Thirty three participants (72%) of the cohort were infected with SARS-CoV-2. We utilized baseline samples from 47 individuals who completed the study, for IgG and IgA antibody profiling. One participant who reported a previous SARS-CoV-2 infection was removed from our analysis. Individuals were ranked by baseline IgG and IgA binding magnitude to the Wuhan strain and to a panel of VOCs. We used the same quartile approach to define low- mid- and high-baseline immune history groups. The same Cox proportional hazards model described above was used. The model was adjusted for all relevant and available covariates including age, sex, and number of vaccine doses. Hazard ratios comparing the infection rates in the low- and high-baseline groups were computed at two timepoints: (1) April 2022—at the end of the BA.1/BA.2 omicron wave in Israel; and (2) June 2022—at the end of the study followup period, which also included additional BA.4/5 infections.

### Statistical analysis

The results are presented as the mean (SD) for continuous variables and as the total patients (percentage of total patients) for categorical data. A t-test was used to compare the continuous variables and chi-square test for categorical data, using Fisher's exact test if needed. In addition, we used Mann–Whitney test to compare variables without normal distribution. In the primary analysis, we compared the rates of Covid-19 infection among different serological response groups. The Kaplan-Meier estimator was used to construct cumulative incidence curves describing the infection rate. We used the Cox proportional hazards model adjusted for age, sex, occupation (physician/nurse or administrative/support staff), medical center, and time from the third vaccination to assess risk of infection. The risk was defined as the fold increase in the hazard of being infected. We used calendar time as the time scale to account for fluctuations in infection rates. We used the same model to assess vaccine efficacy (VE), defined as one minus the hazard ratio. Previous studies demonstrated that VE following a fourth dose of the Pfizer mRNA vaccine wanes after 30 days[11]. In line with these findings, we found that the cumulative incidence curves of the third and fourth dose recipients in our study became parallel around day 30 (Fig. 1f), indicating similar infection rates from this time-point and on. Therefore, we estimated VE at day 30 and also analyzed VE for the entire interim followup time in which all participants were followed for at least 60 days. Since vaccination with a fourth dose may modify VE, we analyzed COP separately in the three and four-dose groups. Furthermore, due to the possible time-limited VE, we analyzed these differences at day 30 and at the interim followup time-point.

To explore the robustness of our estimates, we performed a sensitivity analysis for the main results. We applied a Poisson regression adjusted to the same variables aforementioned and the daily proportion of positive PCR tests. In addition, we added subject IDs as a random effect to account for repeated measures. This analysis defines risk as the fold increase in the incidence rate ratio and VE as 1 minus the incidence rate ratio.

### Ethics

The study was approved by the CHS Central Institutional Review Board (0404-21-SOR-C). The Validation cohort study was approved by the institutional review board of the Soroka University Medical Center (SOR20-0371). All participants provided written informed consent. The report follows the STROBE methodology[58].

### Infectious-virus neutralization assays

Viruses in neutralization assays were isolated from de-identified, discarded nasal swabs and grown in VeroE6 cells ectopically expressing

both TMPRSS2 and human ACE2 (VE6/T2/ACE2; provided by Dr. Barney Graham at VRC, NIAID, NIH). Briefly, 100uL of swab suspension was inoculated onto VE6/T2/ACE2 cells seeded in 6 well tissue culture plates and incubated at 37 °C, 5% $CO_2$ until 90% cytopathic effect (CPE) was observed. The presence of the virus was confirmed by BD Veritor System for rapid detection of SARS-CoV-2 (Catalog # 256082). Virus stocks were subsequently expanded using a VeroE6 cell line ectopically expressing TMPRSS2 (VE6/T2; from JCRB Cell Bank, Japan (https://cellbank.nibiohn.go.jp/english/)). Briefly, VE6/T2 cells were inoculated with the virus at a 1:50 dilution and incubated at 37 °C, 5% $CO_2$ until 90% CPE was observed. Virus stocks were tittered in VE6/T2 cells to determine a 50% tissue culture infectious dose (TCID50). Cells in a 96 well format were inoculated with a 1:10 serially diluted virus stock for 72 h. Wells were stained with 0.1% crystal violet solution to visualize cells. Infectious dose titers were determined using the Reed and Muench method.

Infectious-virus microneutralization assays were performed in a 96-well format. Half-log serial dilutions of heat-inactivated plasma or sera (1 h at 56 °C), starting at a 1:50 dilution, were incubated with 250 $TCID_{50}$ of infectious SARS-CoV-2 virus at a 1:1 ratio for 1 h at 37 °C. The serum/plasma mixture was then added to VE6/T2 cells and incubated at 37 °C, 5% $CO_2$ for 24–48 h. Following incubation, cells were fixed with 4% formaldehyde for 30 min, washed with PBS, and incubated with a block/permeabilization buffer (PBS supplemented with 3% Bovine Serum Albumin and 0.2% Triton-X-100) for 30 min. Rabbit anti-SARS CoV-2 NP mAb (Sino Biologicals Cat # 40143-R040) at a 1:2000 dilution was added for 1 h and cells were washed with PBS supplemented with 0.5% Tween (PBST) before incubation with a secondary goat anti-rabbit IgG–HRP conjugated antibody (Cell Signaling Cat# 7074 S)) at a 1:3000 dilution for 1 h. Finally, cells were washed with PBST and incubated with TMB for 10 min before 1 N sulfuric acid (Fisher Scientific Cat #SA212-1) was added to stop the reaction. The optical density was measured at 450 nm on a Biotek Synergy plate microplate reader. To compute EC50 values, we subtracted the mean of the negative control from all wells, and fitted values using a five parameter logistic regression model (5PL) using the python scipy package. The readout of the positive control was used as the maximal response for curve fitting.

## Pseudovirus neutralization assays

Pseudotyped viruses were generated in HEK293T cells. Pseudoviruses were generated following transfection of, LTR-PGK luciferase lentivector into HEK293T cells together with lentiviral packaging plasmids coding for Gag, Pol Tat Rev, and the corresponding wild type or mutate SARS CoV-2 spike envelopes. Transfections were performed in a 10 cm format and the supernatant containing virus was harvested 72 h post-transfection, filtered, and stored at −80 °C as previously described (Krasnopolsky et al., 2020). Pseudovirus quality control and titers were determined by transducing HEK293T cells expressing ACE2 (HEK-ACE2) that were plated in a 12-well plate. After 24 h, transduction was monitored serial dilutions of pseudovirus were used. After 48 h of post-transduction, cells were harvested and analyzed for their luciferase readouts. p24 ELISA measurements were also conducted to ensure equal loads.

Neutralization assays were performed in a 96 well format, in the presence of pseudotyped viruses that were incubated with increasing dilutions of the tested sera (1:50; 1:250: 1;1250: 1:6250: 1:31250) or without sera as a control. Virus and sera were incubated for 1 h. At 37 °C followed by transduction of HEK-ACE2 cells for an additional 12 h. After 72 h post-transduction, cells were harvested and analyzed for luciferase readouts according to the manufacturer protocol (Promega). Neutralization measurements were performed in triplicates using an automated Tecan liquid handler and readout was used to calculate $NT_{50}$ – 50% inhibitory titers concentration. All experiments were run in technical duplicates or triplicates.

## ELISA assay

Recombinant SARS-CoV-2 proteins purchased from Sino Biological include the full-length spike protein (40589-V08H) and RBD (40592-V08H) from the Wuhan-Hu-1 isolate, the RBD of the B.1.617.2 (Delta) variant (40492-V08H90), and the RBD of the BA.1 (Omicron) variant (40592-V08H121). Expression plasmids for the nucleocapsid (N) protein from the Wuhan-Hu-1 isolate and the RBD of the B.1.1.28 or P.1 (Gamma) variant were obtained from Florian Krammer. Plasmids were transfected into Expi293F cells using an ExpiFectamine 293 transfection kit (Thermo Fisher Scientific, A14524) as previously described (Amanat, F. et al. PMID32398876). Supernatants from transfected cells were harvested and purified with a Ni-NTA column. The resulting purified proteins were used for ELISA analysis of serum samples.

For antibody detection by ELISA, 384-well microtiter plates were coated overnight at 4° with recombinant proteins diluted in PBS. Optimal concentrations for each protein and isotype were empirically determined to optimize sensitivity and specificity. The N protein was coated at 1 μg/mL for IgG detection and 2 μg/mL for IgA detection. The full-length spike protein was coated at 2 μg/mL for IgG detection and 4 μg/mL for IgA detection. All RBD proteins were coated at 4 μg/mL for IgG and IgA detection. The following day plates were washed three times with 0.1% PBS-T (0.1% Tween-20) and blocked with 3% OmniblokTM non-fat milk (AmericanBio; AB10109-01000) in PBS-T for 1 h. Plates were washed three times with 0.1% PBS-T immediately before the addition of diluted samples. Prior to dilution, plasma or serum samples were incubated at 56 °C for 15 min and then diluted in 1% milk in PBS-T. Diluted samples were added to the blocked plates and incubated for 90 min at room temperature. The plates were washed three times and incubated for 30 min at room temperature with secondary antibodies diluted in 1% milk in PBS-T: anti-IgG (1:10,000; Invitrogen, A18805) or anti-IgA (1:2,000; Southern Biotech, 2050-05). The plates were washed and incubated at room temperature with SIGMAFAST OPD (Sigma-Aldrich; P9187) for eight min. The chemiluminescence reaction was stopped by the addition of 3 N HCl and absorbances were measured at 490 nm on a microplate reader. To control for plate-to-plate variability, the same positive and negative control samples are included on each plate. In addition, the WHO international standard from the National Institute of Biological Standards and Control (NIBSC, cat# 21/234) was included on each plate. The WHO standard contained 817, 832, and 713 binding antibody units (BAU)/mL for the RBD, full-length spike, and N IgG, respectively. For the IgA, we previously calculated the BAU/mL of our control samples using the NIBSC standard 20/136, which was 1000 BAU/mL for all antigens and isotypes. All OD values were converted to BAU/mL using the reference standards on each plate.

## Antigen microarray spotting

Recombinant SARS-CoV-2 proteins were spotted onto N-hydroxysuccinimide ester–derivatized Hydrogel slides (H slides) using a Scienion Sx non-contact array spotter. The proteins were purchased from Sino Biological (China) or were obtained as gifts through BEI Resources (NIAID, NIH), from ACROBiosystems, as listed in Supplementary Data 5. Each recombinant SARS-CoV-2 protein was diluted in PBS to the concentration of 130 μg/mL and was spotted in 3 concentrations (65, 35, and 16.25 μg/mL) in 0.0025% Triton X-100. Spot volumes ranged between 300 and 360 pL. Each antigen at each concentration was spotted in triplicate. Sixteen identical microarrays were spotted on each microarray slide. All samples were profiled using microarrays from a single printing batch, which included 140 microarray slides each containing 16 arrays per slide.

## Antigen microarray assay

Array slides were blocked with 4 mL chemical blocking solution per slide (50 mM ethanolamine, 50 mM borate, pH 9.0) for 1 h at room temperature (RT) on a shaker. After blocking the liquid was vacuumed,

the slide was washed 2 times for 3 min in a washing buffer (0.05% tween-20 in PBS), 2 times for 3 min in PBS and an additional 3 min wash in double deionized water (DDW). Every wash was with 3 mL of liquid per slide on a shaker at RT. Samples were diluted in a hybridization buffer (1% BSA/0.025% tween-20 in PBS). Human serum samples were diluted at 1:1000 for IgG characterization and 1:100 for IgA characterization. Following 2 h of incubation, the slides were dried by centrifugation at RT for 5 min at a speed of 800 g in a slide holder padded with Kim wipes, loaded on divided incubation trays (PepperChips, PepperPrint, Germany), and then the samples were added and hybridized with the arrays for 2 h at RT on a shaker. After hybridization, the samples were discarded and the slides were washed twice with a washing buffer and twice with PBS as described above. After washes, the slides were incubated for 45 min on the shaker at RT with a fluorescently labeled polyclonal secondary antibody in the hybridization buffer. The secondary antibody for IgG was Alexa Fluor® 647 affinipure Donkey Anti-Human IgG (H + L), cat# 709-605-149, Jackson ImmunoResearch at 1:1000 dilution. The secondary antibody for IgA was Alexa Fluor® 647 affinipure Goat Anti-Human serum IgA a Chain Specific, cat# 109-605-011, Jackson ImmunoResearch at 1:5000 dilution. To detect bound immunoglobulins, slides were scanned on a three-laser GenePix 4400 scanner. Images were analyzed using GenePix Pro version 7 to obtain the mean fluorescence intensity (MFI) of each spot after subtracting the mean local background fluorescence intensity ($0 \leq MFI \leq 65{,}000$).

### Antigen microarray analysis

The array results were uploaded to a python pandas dataframe and analyzed using python scripts. Since each antigen at each concentration was spotted in triplicate, the median fluorescent intensity (MFI) of each triplicate was calculated. During each experiment, a negative control array was hybridized with the hybridization buffer only. The background staining of the negative control array was subtracted from each other array. Since the antigens were spotted in serial concentrations, a 5-parameter logistic regression model was used to fit curves to the measured MFI across all antigen concentrations, and the area under the curve (AUC) was calculated for each antigen. The magnitude of antibody response to a group of antigens was defined as the sum of MFI AUC of all the proteins included in the group. We summed 3 groups of antigens for magnitude computation: 1. Wuhan - Wuhan spike and RBD antigens (whole S1 + S1 protein, S1 subunit alone, RBD alone); 2. Variants - whole spike antigens (S + S2) of non-Wuhan SARS-CoV-2 variants: B.1.1.1, A.23.1, Alpha, Beta, Gamma, Delta, AY.2, Iota, Kappa, Mu, Theta and R.1; 3. RBD Mutants - Wuhan RBD sequence including specific mutation: V483A, K417N/E484K/N501Y, L452R, or N440K. We used mean centering[59] to normalize results across different experiments.

### Ranking participants by baseline markers

We used various different baseline markers to rank participants from highest to lowest. First, rankings were computed within the three-dose and four-dose groups separately. Then, after ranking participants, we used a quartile analysis to define three baseline groups: (1) 'low' - lowest quartile; (2) 'mid' - quartiles 2 and 3; and (3) 'high' - highest quartile.

For the two commercially available serological assays tested, we also used a single threshold approach to divide participants into a low baseline and high-baseline group as follows: (1) BioPlex 2200 SARS-CoV-2 IgG panel, S2 (Biorad Laboratories, Cal, USA) - we used the clinical cutoff provided by the manufacturer (<10) to define the 'low' group, and all individuals with a titer >= 10 were defined as 'high'; (2) SARS-CoV-2 IgG II Quant RBD of S1 subunit (Alinity, Abbott, USA): Since all participants had a measurable titer at baseline, we used the median titer (4560) as a single threshold. Participants with titers <4560 were

assigned to the low-baseline group and participants with titer >= 4560 were defined as the high-baseline group.

### Combining clinical markers

To combine the rankings of two serological baseline markers, we considered the intersection of the two low-baseline and high-baseline groups to define the low and high groups of each combination. We only considered combinations of markers that were significantly associated with protection for either three-dose or four-dose individuals, considering both the 30-day and interim followup timepoints.

### Immunogenicity subset selection

We used baseline responses of participants to rank them based on their responses to the S1 and RBD antigens of the Wuhan wildtype strain. We selected 38 participants with low baseline antibody levels (low-baseline) and 36 participants with high antibody levels (high-baseline) for an in-depth immunogenicity assessment at the baseline and day 30 post-vaccination time points (Supplementary Data 3-4). Participants were selected at enrollment and included 58 (42.92%) 4th dose recipients. 23 (17.02%) participants were infected within the first 30 days of the trial (Supplementary Data 4).

### Reporting summary

Further information on research design is available in the Nature Portfolio Reporting Summary linked to this article.

## Data availability

All data needed to evaluate the conclusions in the paper are present in the paper or the Supplementary Data and source data file. Source data are provided with this paper.

## Code availability

All analysis code is available in GitHub (https://github.com/shllevy/COP.git).

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

## Acknowledgements

We would like to acknowledge the following for their contribution to this study: Lihi Marciano, Ruthie Bechor, Hana Kahanov-Edelstein, Galit Carmon, Moran Raad, and Natalie Samiande. The clinical study was funded by Clalit Health Services and the Clinical Research Center at the Soroka University Medical Center. T.H. was supported by the Israel Science Foundation (ISF) grants no. 882/17 and 2683/21; and NIH award no. 75N93021C00016 subaward# 11308501A-8028280. R.T. is supported by the Israeli Ministry of Science and Technology f (MOST; grant #3-16897), the Israel Science Foundation for R.T. (ISF; Research Grant Application no. 755/17) and the Ben-Gurion University of the Negev COVID-19 Research Task Force for R.T. Richard Webby was funded by the Federal funds from the National Institute of Allergy and Infectious Diseases, National Institutes of Health, Department of Health and Human Services, under Contract No. 75N93021C00016".

## Author contributions

T.H., V.N., R.T., L.N.E., O.W. designed the study; Y.S.A, G.W., B.C., M.C., contributed to the methodology; G.W., R.N.D., B.C., V.N., and L.N.E. were involved in clinical recruitment and clinical study; S.L., H.O., S.Z., A.K., L.M.F., S.T., D.B., L.C.Y., O.V., carried out experiments; T.H., R.T., R.W., M.A.M., and Y.S.A. designed and supervised experiments. D.A., M.L.C., and N.K.A. provided laboratory support, L.N. and V.N. provided statistical guidance; T.H., S.L., D.O., H.O., L.C.L., L.M.F., Y.S.A., L.N., V.N., R.T., and L.N.E. were involved with statistical analysis and data interpretation. T.H., S.L., D.O., R.T., L.N.E., drafted the manuscript; T.H., Y.S.A., M.A.M., R.W., M.C., R.T., L.N.E. critically revised the manuscript; all authors approved the final version for publication.

## Competing interests

The authors declare no competing interests.

## Additional information

[1]Department of Microbiology, Immunology and Genetics, Faculty of Health Sciences, Ben-Gurion University of the Negev, Beer-Sheva, Israel. [2]National Institute of Biotechnology in the Negev, Ben-Gurion University of the Negev, Beer-Sheva, Israel. [3]Vaccine and Infectious Disease Division, Fred Hutch Cancer Research Center, Seattle, USA. [4]Clinical Research Center, Soroka University Medical Center, and the faculty of Health Sciences, Ben-Gurion University of the Negev, Beer-Sheva, Israel. [5]Department of Infectious Diseases, St. Jude Children's Research Hospital, Memphis, TN, USA. [6]Department of Immunology, St. Jude Children's Research Hospital, Memphis, TN, USA. [7]Laboratory of Virology, Soroka University Medical Center, Beer-Sheva, Israel. [8]Laboratory of Management, Soroka University Medical Center, Beer-Sheva, Israel. [9]Central Laboratory, Clalit Health Services & Dept. of Clinical Biochemistry and Pharmacology, Faculty of Health Sciences, Ben-Gurion University of the Negev, Beer Sheba, Israel. [10]Infectious Diseases Unit, Lady Davis Carmel Medical Center, Haifa, Israel. [11]Rappaport Faculty of Medicine, Technion-Israel Institute of Technology, Haifa, Israel. [12]Infectious Diseases Unit, Emek Medical Center, Afula, Israel. [13]School of Medicine, Tel Aviv University, Tel Aviv, Israel. [14]Meir Medical Center, Kfar Saba, Israel. [15]Infectious Disease Institute, Soroka University Medical Center, and Faculty of Health Sciences, Ben-Gurion University, Beer Sheba, Israel. [16]Dept. of Health systems management, faculty of health sciences, Ben-Gurion University of the Negev, Beer Sheva, Israel. [17]Hospital division, Clalit Health Services, Tel Aviv, Israel. [18]These authors contributed equally: Shlomia Levy, Daniel Ostrovsky, Hannah Oppenheimer, Shosh Zismanov, Alona Kuzmina. [19]These authors jointly supervised this work: Tomer Hertz, Ran Taube, Lior Nesher. ✉e-mail: thertz@post.bgu.ac.il; rantaube@bgu.ac.il; nesherke@bgu.ac.il

