## [Peer Review File · Nature Communications]

Correlates of protection for booster doses of the SARS-CoV-2 vaccine BNT162b2Editorial Note: This manuscript has been previously reviewed at another journal that is not operating a transparent peer review scheme. This document only contains reviewer comments and rebuttal letters for versions considered at *Nature Communications*.

REVIEWER COMMENTS

Reviewer #1 (Remarks to the Author):

The authors of the manuscript should be commended as they have addressed most of the comments made in my initial review. Some issues still need to be addressed however

- The accepted way to validate biomarkers is to perform an ROC (receiver operating characteristic curve) . In this analysis the biomarker identified in the training cohort is tested for its capacity to predict the accuracy of these biomarkers on an independent validation cohort. The authors should consider performing a ROC curve analysis which will confirm the accuracy of their biomarkers.
- ROC Curve analysis on each of the biomarkers identified (IgA , IgG titers , crossreactivity with variant of concerns) will enable the authors to stratify and rank the different biomarkers they identified based on statistically valid models .
- Finally a multivariate analysis will enable rigorous assessment of the independent predictive values of each of the biomarkers identified . These additional measures would considerably strengthen the manuscript .

While the authors did not provide any mechanisms that can explain the dichotomy of subjects that show high or low levels of SARS COV2 specific Abs, they could suggest in their discussion narrative on possible mechanisms that can explain such differences. The role of baseline immunological and host features on the response to vaccines has been the subject of several publications in the recent literature

Reviewer #2 (Remarks to the Author):

In this manuscript, Hertz et al show that baseline antibody markers are associated with infection risk and can be utilized to identify individuals at risk from future exposures. This manuscript describes a valuable dataset that could contribute to our understanding of correlates of protection against SARS-CoV-2 infection, but in its current form I find the manuscript extremely difficult to read, and I find it almost impossible to follow the authors' reasoning.

Major comments

- The abstract puts a major focus on the comparison between 3- or 4-doses. However, it is unclear from the abstract when samples were obtained, making it impossible to understand the experiments and the author line of reasoning. The authors are unclear on their hypothesis and research question. Line 55-56, without introduction, it is strange to say that 'ELISA assays were associated with protection'. The authors mean IgG or IgA levels, I assume?

- Line 62-63: 'we identify a highly susceptible population that remains susceptible despite apparent responsiveness to vaccines.' How is responsiveness defined, the ability to form antibodies? Does this not suggest that other immunological parameters are important?

- The introduction does not set the stage for the paper. The authors do not introduce what is known about correlates of protection, the different immunological parameters (what about cellular immunity) potentially involved in protection, the difference in protection from infection or disease, etc. Rather, the authors use most of the introduction as a methods section, from line 88 onwards. This needs to be completely redrafted.

- I think the authors have really done a poor job describing their sample sets. They collected samples from donors vaccinated three of four times, and collected a baseline and day 30 sample. I assume that the donors that received a fourth vaccination were vaccinated at baseline, but this was not the case for the donors that received a third vaccination. These were just to samples with a 30-day interval without intervention? After reading the abstract and results section multiple times this is still not clear to me, and the description should be improved. In any event, in line 132-133 the authors state that 'no significant titer rises were observed in individuals who only received three doses.' This was not expected I assume, why do the authors highlight this?

- Related to this: the authors find more infections in the three-dose group compared to the four dose group. However, as the third vaccine has been delivered ago, there must have been a significant difference in the time interval between previous vaccination and infection. Concretely: is the difference in infection rate a function of the number of vaccines administered, or time since last vaccination?

- The neutralization data presented in the manuscript is not novel (up to BA.2, already shown by others). Additionally, neutralization is not a parameter assessed by the authors in their CoP analysis. What does the neutralization data add to this manuscript?

- It would be helpful if the authors provide correlations between the different measurements. In line 232-234 the authors state that combinations of baseline markers were significantly associated with infection risk, which makes sense of the immunological parameters were correlated.

- In many instances in the results (and figures) the authors state to have measured antibodies to 'variants' or 'mutants'. Which variant? What mutant? This is completely unclear.

- Line 272: the authors claim that the vaccine induced antibodies that were correlated with protection, yet 30% of the included participants became infected. Was information on disease severity collected?

- The manuscript has a fairly long discussion, but limited description of the implications of the authors' findings. Should we do antibody level testing to determine who needs revaccination? It would be nice to add implications to the last paragraph of the discussion.

Minor comments

- In the abstract the authors state that VOC pose a serious global problem. Some might argue that VOC from the omicron sub-lineage are mild, place in perspective.

- Terminology is important and should be corrected, for example 'SARS-CoV-2 virus' or 'infectiveness', which should be 'SARS-CoV-2' and 'infectivity'. The use of the terminology VOC is a bit outdated. Line 131-132: the authors measured RBD ELISA levels, a rise in titer against 'SARS-CoV-2 isolates' is speculative. Rather use 'infectious' virus over 'live' virus. Line 226-227: 'combinations of baseline markers improved COPs in both groups'? What does this mean?

- As the authors have longitudinal samples taken with short intervals, did they consider including nucleocapsid-specific antibodies as a measure for (prior) infections.

- Line 107: do the authors mean '365 received three doses OF WHICH 165 were infected'?

- The authors find discrepancies between their infectious virus and pseudovirus neutralization assay. Explain!

- In line 165-167 the authors describe high-baseline and low-baseline responders, and refer to figure 2A. This is again confusing, as that figure also shows medium responders and the numbers don't seem to add up.

- It is also a bit weird that figure 2A is described twice with a different message (line 165-176 and line 197-199). This does not add to the clarity of the manuscript. Individuals were divided into quartiles defining low, mid and high. What's the fourth quartile?

- What is the added value of the spider plots in figure 1 and 2. The authors poorly describe these in the results section, and in my opinion, they are unnecessarily complex.

- Line 185-187: are these differences expected (why are antibody levels lower in prior infected), and are they relevant as the ranges almost completely overlap?

- Line 211-212: Example of difficult to read result. What does the Abbott Alinity measure? It is not helpful of different terminology is used, and readers need to refer to the methods section to understand.

Reviewer #1 (Remarks to the Author):

The authors of the manuscript should be commended as they have addressed most of the comments made in my initial review. Some issues still need to be addressed however

- The accepted way to validate biomarkers is to perform an ROC (receiver operating characteristic curve) . In this analysis the biomarker identified in the training cohort is tested for its capacity to predict the accuracy of these biomarkers on an independent validation cohort. The authors should consider performing a ROC curve analysis which will confirm the accuracy of their biomarkers.*
- ROC Curve analysis on each of the biomarkers identified (IgA , IgG titers , crossreactivity with variant of concerns) will enable the authors to stratify and rank the different biomarkers they identified based on statistically valid models .*
- Finally a multivariate analysis will enable rigorous assessment of the independent predictive values of each of the biomarkers identified . These additional measures would considerably strengthen the manuscript .*

Thank you for your insightful comment. We agree that one of the most common ways to train and validate biomarkers is to use classifier metrics such as ROC. However, for the current study we believe that survival analysis is more suitable. One of the main challenges when conducting analysis on this cohort is the different follow-up times for each participant, survival analysis unlike ROC has the ability to accommodate different follow-up times. Additionally, we compared the infection rates between the upper and lower quartiles for each marker. When analyzing only two quartiles, using an ROC analysis could be misleading. Nevertheless, we believe that performing a ROC analysis would produce similar results for ranking markers by their prognostic performance. To demonstrate this, we conducted an AUC analysis using logistic regression to predict infection at the 30-day timepoint, which is the longest follow-up time shared by all participants. As expected, we found that the hazard ratio magnitudes and the AUCs are ordered similarly.

4 doses

Single	AUC	HR
IgA Variants	0.70	4.45
IgA Wuhan	0.68	3.19
IgG Mutants RBD	0.66	2.07
IgG Alinity RBD	0.65	1.47

IgG BioPlex S2	0.64	1.32
Combinations		
IgA Variants & IgG Mutants RBD	0.72	12.24
IgA Wuhan & IgG Mutants RBD	0.71	6.15
IgA Variants & IgG BioPlex S2	0.71	3.04
IgA Variants & IgG Alinity RBD	0.71	4.38
IgA Variants & IgA Wuhan	0.70	5.73
IgA Wuhan & IgG Alinity RBD	0.70	2.51
IgA Wuhan & IgG BioPlex S2	0.69	2.87
IgG Mutants RBD & IgG BioPlex S2	0.67	2.77
IgG Mutants RBD & IgG Alinity RBD	0.67	1.96
IgG BioPlex S2 & IgG Alinity RBD	0.65	1.72

3 doses

Single	AUC	HR
IgG Alinity RBD	0.65	1.59
IgG BioPlex S2	0.64	1.39
IgG Mutants RBD	0.63	1.5
IgA Variants	0.62	1.34
IgA Wuhan	0.62	1.09

Combinations		
IgA Variants & IgG Mutants RBD	0.66	4.49
IgA Wuhan & IgG Alinity RBD	0.66	1.64
IgG Mutants RBD & IgG Alinity RBD	0.66	1.29
IgA Variants & IgG Alinity RBD	0.65	3.45
IgG BioPlex S2 & IgG Alinity RBD	0.65	1.76
IgA Wuhan & IgG Mutants RBD	0.64	2.75
IgG Mutants RBD & IgG BioPlex S2	0.64	1.93
IgA Wuhan & IgG BioPlex S2	0.64	1.63
IgA Variants & IgG BioPlex S2	0.63	2.94
IgA Variants & IgA Wuhan	0.62	1.48

Validation Cohort

Single	AUC	HR
IgG Variants	0.71	3.56
IgG Wuhan	0.66	3.96
IgA Wuhan	0.62	2.02
IgA Variants	0.62	1.62
Combinations		

IgG Wuhan & IgA Variants	0.75	5.05
IgG Variants & IgA Variants	0.74	4.14
IgG Variants & IgA Wuhan	0.72	8.50
IgG Wuhan & IgA Wuhan	0.69	2.96

While the authors did not provide any mechanisms that can explain the dichotomy of subjects that show high or low levels of SARS COV2 specific Abs, they could suggest in their discussion narrative on possible mechanisms that can explain such differences. The role of baseline immunological and host features on the response to vaccines has been the subject of several publications in the recent literature

We thank the reviewer for this comment. We have revised the discussion to include the following:

Other recent studies have highlighted the role of baseline immunological and host features on the response to vaccines ³⁵⁻³⁷.

We also added the following text regarding potential mechanisms:

Additional studies are required to assess whether this subpopulation is also at an increased risk for severe disease, and whether it may spread infection more readily than others. While the underlying mechanism for the increased susceptibility to symptomatic infection in this subpopulation is currently unknown, our study found that these individuals are indeed capable of mounting neutralizing antibody titers following an additional booster shot, suggesting that other functional differences between these groups such as Fc effector functions and antibody waning dynamics may be at play. These findings warrant further longitudinal functional studies of this group across longer followup time.

Reviewer #2 (Remarks to the Author):

In this manuscript, Hertz et al show that baseline antibody markers are associated with infection risk and can be utilized to identify individuals at risk from future exposures. This manuscript describes a valuable dataset that could contribute to our understanding of correlates of protection against SARS-CoV-2 infection, but in its current form I find the manuscript extremely difficult to read, and I find it almost impossible to follow the authors' reasoning.

Major comments

- *The abstract puts a major focus on the comparison between 3- or 4-doses. However, it is unclear from the abstract when samples were obtained, making it impossible to understand the experiments and the author line of reasoning.*

We thank the reviewer for this comment. We have extensively revised the abstract and manuscript to make it more readable and easier to follow and specifically clarified that all individuals in the study previously received 3 doses of the vaccine, and that a subset of them opted to receive a fourth dose at enrollment.

- *The authors are unclear on their hypothesis and research question.*

We have rewritten the introduction of the paper following several of your comments and in particular the revised introduction now clearly states our hypothesis and research question. Below is the revised last paragraph of the introduction:

Here we report an interim analysis of the Clalit HCPs Booster study - a multicenter prospective trial in healthcare providers with increased risk of SARS-CoV-2 infection designed to identify novel correlates of protection (COP) for booster doses of the Pfizer-BioNTech vaccine. We hypothesized that immune history to SARS-CoV-2, including both the Wuhan wildtype strain and multiple variants of concern, would be associated with infection risk with the Omicron VOC. To represent immune history, we measured the IgG and IgA serum magnitude of antibody responses to the spike and RBD proteins of multiple SARS-CoV-2 VOCs and their combinations as correlates of protection (COPs). In addition to research assays, we utilized commercial assays that are in clinical use and can be widely used to identify individuals with an increased risk of SARS-CoV-2 infection that may benefit from passive immunization and have clinical and epidemiological implications.

- *Line 55-56, without introduction, it is strange to say that 'ELISA assays were associated with protection'. The authors mean IgG or IgA levels, I assume?*

We agree. We have revised the sentence as follows:

A combination of IgG levels measured using two commercially available ELISA assays was also associated with protection in both groups (HR = 1.84, p = 0.002; HR = 2.01, p = 0.025, respectively).

- *Line 62-63: 'we identify a highly susceptible population that remains susceptible despite apparent responsiveness to vaccines.' How is responsiveness defined, the ability to form antibodies? Does this not suggest that other immunological parameters are important?*

The previous sentence states that:

Most importantly, we identified a subset of individuals with low antibody levels after three doses of vaccine that responded with a significant boost in neutralizing antibody titers after a fourth dose, but were still at significantly increased susceptibility to infection when compared to those who had pre-existing high levels of binding antibodies.

This is the responsiveness the sentence refers to. We agree that this suggests that other immunological parameters may indeed be important for protection, and we have added text in the discussion following additional comments made below.

- The introduction does not set the stage for the paper. The authors do not introduce what is known about correlates of protection, the different immunological parameters (what about cellular immunity) potentially involved in protection, the difference in protection from infection or disease, etc. Rather, the authors use most of the introduction as a methods section, from line 88 onwards. This needs to be completely redrafted.

We thank the reviewer for this important comment. We have rewritten the introduction and it now includes a paragraph that discusses what is known on correlates of protection:

Correlates of protection are immune markers that can be used to predict vaccine efficacy against infection or disease after vaccination¹⁴⁻¹⁷. Neutralizing antibodies or binding antibodies have been established as a correlate of protection for vaccines against many viral diseases¹⁵. More specifically, recent studies have shown that neutralizing antibody titers and IgG binding titers to the SARS-Cov-2 spike protein are correlates of protection from symptomatic infection following vaccination with mRNA vaccines and the ChadOx AstraZeneca vaccine^{16,18-21}. Additional studies have highlighted the role of cellular responses as correlates of protection and in reducing disease severity²²⁻²⁵.

- I think the authors have really done a poor job describing their sample sets. They collected samples from donors vaccinated three of four times, and collected a baseline and day 30 sample. I assume that the donors that received a fourth vaccination were vaccinated at baseline, but this was not the case for the donors that received a third vaccination. These were just to samples with a 30-day interval without intervention? After reading the abstract and results section multiple times this is still not clear to me, and the description should be improved.

We once more thank the reviewer for this important comment. We have modified the text in multiple places to clarify our sample sets (abstract, introduction and methods). All of the 607 study participants received three doses as part of the standard 3 dose regimen recommended by the Israeli MOH, which included 2 doses 21 days apart and a third dose 6 months after the 2nd dose. Upon enrollment, a subset of 242 opted to receive a fourth dose at enrollment.

In any event, in line 132-133 the authors state that 'no significant titer rises were observed in individuals who only received three doses.' This was not expected I assume, why do the authors highlight this?

We have modified the sentence to clarify it refers to participants that received 3 doses and were un-infected within the first 30 days as follows:

No significant rises were observed in uninfected individuals who only received three doses.

- Related to this: the authors find more infections in the three-dose group compared to the four dose group. However, as the third vaccine has been delivered ago, there must have been a significant difference in the time interval between previous vaccination and infection. Concretely: is the difference in infection rate a function of the number of vaccines administered, or time since last vaccination?

The question raised by the reviewer is clearly of great interest, but is a difficult one to address in the interim analysis presented here. To clearly address the question one would have to compare individuals with 3 or 4

doses at a similar interval post-boost, which would require comparing infection rates at different dates and probably from different variants. This would further be complicated by the change in attack rates at different points of time. This type of analysis is outside the scope of this manuscript, thus we would not be able to address this question adequately. However, we have observed in our analysis that the additional protection from infection provided by the 4th dose was relatively short lived, with infection rates evening out after about 30 days. Based on this observation, we suggest that additional booster shots should be administered early during the onset of a new wave, where they would be important to reduce the spread of a new variant.

We have added the following text to the discussion of the paper:

This rise in binding and neutralizing antibody titers was also associated with increased protection against symptomatic Omicron B.A1 infection, but this effect was transient. This suggests that the additional boosters should be administered at the onset of new infection waves where they may be important for reducing the spread of a new variant.

- The neutralization data presented in the manuscript is not novel (up to BA.2, already shown by others). Additionally, neutralization is not a parameter assessed by the authors in their CoP analysis. What does the neutralization data add to this manuscript?

The neutralization data presented here up to BA.2 was indeed shown by others, but its purpose in our manuscript was to allow us to functionally compare the individuals in the low-BIH and high-BIH groups. As we note in the paper, we did not just measure neutralization for a random subset of individuals from the study, but rather screened individuals at baseline and selected a subset of 76 individuals from the two ends of the BIH spectra for further in-depth immune characterization. Importantly, as stated in the abstract, while we find significant differences between the neutralization titers of the low- and high-BIH groups at baseline, these differences are no longer significant at day 30 post vaccination. This suggests that the differences between the low- and high-BIH group are not due to lack of ability to respond to vaccination.

- It would be helpful if the authors provide correlations between the different measurements. In line 232-234 the authors state that combinations of baseline markers were significantly associated with infection risk, which makes sense of the immunological parameters were correlated.

In Supplementary Figure 3 - we plotted a subset of the pairwise correlations between IgG and IgA markers and showed the correlations were relatively weak ($0.08 < r < 0.31$). These weak correlations motivated us to consider combinations of markers, since if the correlations were high, their combination would not modify baseline rankings.

- In many instances in the results (and figures) the authors state to have measured antibodies to 'variants' or 'mutants'. Which variant? What mutant? This is completely unclear.

We apologize for the lack of clarity on this matter. We have revised the text to clearly state which spike variants are included within this set as follows:

Our arrays included spike antigens from the Alpha, Beta, Wuhan, Delta, Gamma, Iota, Kappa, Mu, Theta VOCs and multiple Wuhan spike antigens containing point mutations (**Supplementary Table 1**).

- Line 272: the authors claim that the vaccine induced antibodies that were correlated with protection, yet 30% of the included participants became infected. Was information on disease severity collected?

Information on disease severity was indeed collected, but since all participants previously received 3 doses of Pfizer and were healthy adults without any known medical conditions, we did not observe any severe disease in our cohort.

- The manuscript has a fairly long discussion, but limited description of the implications of the authors' findings. Should we do antibody level testing to determine who needs revaccination? It would be nice to add implications to the last paragraph of the discussion.

We agree. We have now modified the last paragraph of the discussion as follows:

In conclusion, our study demonstrated that combinations of IgA and IgG baseline antibody levels to SARS-CoV-2 VOCs are associated with protection from symptomatic infection. Importantly, our study identified a subpopulation of healthy adult individuals with low-baseline levels of IgA and IgG who are at increased risk for SARS-CoV-2 infection, despite receiving three or four doses of the Pfizer-BioNTech vaccine. Additional studies are required to assess whether this subpopulation is also at an increased risk for severe disease, and whether it may spread infection more readily than others. While the underlying mechanism for the increased susceptibility to symptomatic infection in this subpopulation is currently unknown, our study found that these individuals are indeed capable of mounting neutralizing antibody titers following an additional booster shot, suggesting that other functional differences between these groups such as Fc effector functions and antibody waning dynamics may be at play. These findings warrant further longitudinal functional studies of this group across longer followup time.

Minor comments

- In the abstract the authors state that VOC pose a serious global problem. Some might argue that VOC from the omicron sub-lineage are mild, place in perspective.

It is unclear whether circulating Omicron variants such as XBB are indeed milder, and since this is not the focus of our paper, we don't think it would make sense to address this issue in the paper abstract.

- Terminology is important and should be corrected, for example 'SARS-CoV-2 virus' or 'infectiveness', which should be 'SARS-CoV-2' and 'infectivity'. The use of the terminology VOC is a bit outdated. Line 131-132: the authors measured RBD ELISA levels, a rise in titer against 'SARS-CoV-2 isolates' is speculative. Rather use 'infectious' virus over 'live' virus.

We thank the reviewer for his comments on terminology. We have now revised the paper and modified specific terms noted above.

Line 226-227: 'combinations of baseline markers improved COPs in both groups'? What does this mean?

We have revised the sentence as follows:

We found that combinations of baseline markers were more strongly associated with infection risk in both the three dose and four dose groups than single baseline markers (**Fig. 5, Supplementary Table 11-12**).

- As the authors have longitudinal samples taken with short intervals, did they consider including nucleocapsid-specific antibodies as a measure for (prior) infections.

Indeed we measured nucleocapsid Abs at both timepoints using the commercial Rad BioPlex assay. When plotting these, we could indeed identify several individuals at T1 that had not reported symptomatic SARS-CoV-2 infection, but had detectable levels of NC antibodies, suggesting that they were asymptotically infected during the study. Our study focused on symptomatic infections, and we identified COPs of symptomatic infection. Importantly, multiple studies have reported that not all infected individuals develop NC specific antibodies, and that in many cases these wane over time (Krutikov et al. 2022; Brlić et al. 2022; Swartz et al. 2023). Therefore, NC antibody levels cannot be used to identify all individuals who were previously naturally infected. This was also observed in our study.

We have now added a supplementary figure presenting the NC antibody responses in the un-infected and infected individuals at day 30 (**Supplementary Fig. 5**). We have also added the following text to the manuscript

Our study identified COPs for symptomatic SARS-CoV-2 infection. However, there is significant evidence of frequent asymptomatic SARS-CoV-2 infections. To assess the extent of such infections in our cohort, we analyzed anti SARS-CoV-2 nucleocapsid antibody levels as measured using the Rad-BioPlex assay (Supplementary Fig. 5). We found that 2.9% of the uninfected individuals had detectable NC antibody levels at day 30, suggesting that indeed there were additional asymptomatic infections in our cohort. However, many recent studies have outlined that not all natural infections induce anti-NC antibodies 45,46 and that they also wane quite rapidly in some individuals 47–49, suggesting that not all asymptomatic infections can be detected using anti-NC antibodies.

- Line 107: do the authors mean '365 received three doses OF WHICH 165 were infected'?

We have modified this sentence as follows;

All participants previously received a primary vaccine series of two doses and a third dose six months later. The median number of days from the third vaccination to enrollment was 147. Of the 607 individuals enrolled, 242 (40%) were vaccinated with a fourth dose, of which 74 (30%) became infected, and of the 365 (60%) that did not receive a fourth dose, 165 (45%) were infected (**Table 1**)

- The authors find discrepancies between their infectious virus and pseudovirus neutralization assay. Explain!

Multiple studies have compared SARS-CoV-2 infection-virus (live) and pseudovirus neutralization assays. Overall, studies show that the pseudovirus assays tend to be more sensitive and report higher neutralization titers than live-virus assays (Garcia-Beltran et al. 2022), in line with what we found. We also note that pseudovirus responses were measured only in a subset of 40 individuals from the immunogenicity subset.

- In line 165-167 the authors describe high-baseline and low-baseline responders, and refer to figure 2A. This is again confusing, as that figure also shows medium responders and the numbers don't seem to add up.

Figure 2A represents an example of high-medium-low baseline response group for SARS-CoV-2 Wuhan antigens. The group in the cohort that received the fourth dose of the vaccine (242 individuals) was divided to quartiles while the high quartile of responses belong to the high group and the low quartile belong to the low group and the two quartiles in the middle represent the medium group. Therefore the reference to figure 2A aimed to explain the division to the response group and show the distribution of high-baseline and low-baseline responders.

- It is also a bit weird that figure 2A is described twice with a different message (line 165-176 and line 197-199). This does not add to the clarity of the manuscript. Individuals were divided into quartiles defining low, mid and high. What's the fourth quartile?

We agree. We have completely revised the text in this section to make it more readable.

- What is the added value of the spider plots in figure 1 and 2. The authors poorly describe these in the results section, and in my opinion, they are unnecessarily complex.

Spider plots are commonly used to present the breadth and magnitude of the antibody response to multiple related antigens, allowing to capture a repertoire in a single figure. In figures 1 and 2 we used the average responses of different groups (3 and 4 doses in Figure 1, and low-BIH and high-BIH in Figure 2).

In figure 1 we can nicely see the waning of the response across all antigens in the group that did not receive the 4th dose, and a rise in titer in the group that received a boost. We can also see that the IgG repertoire primarily expands towards Wuhan antigens, and slightly shrinks to other SARS-CoV-2 VOCs.

In figure 2 we can see how significantly different the low-BIH and high-BIH groups are not just in terms of magnitude, but also in terms of breadth. We can also see how the IgG and IgA profiles are not correlated one to another.

We have now revised the figure legends and text to better reflect the points raised above as follows:

Fig. 2: Ranking individuals using baseline binding antibody markers is associated with baseline neutralizing titers

(a) Ranking of 242 vaccinated individuals by their magnitude to SARS-CoV-2 Wuhan at enrollment. Each bar represents the magnitude of a single participant defined as the average response to the set Wuhan antigens (see **Supplementary Table 1**). Participants were divided into low (lowest quartile); mid (quartiles 2 + 3); and high (highest quartile) based on magnitude of IgA responses to Wuhan. (b) Spider plots of the average normalized responses in the low-BIH and high-BIH groups to a set of spike and RBD proteins including the Wuhan spike and RBD, RBD mutants and multiple SARS-CoV-2 variants of concern spike proteins. Responses of the low and high response groups of 127 uninfected individuals that received a fourth boost are plotted separately for IgA (top) and IgG (bottom). (c) Live-virus neutralization titers of 45 vaccinated uninfected individuals at day 0 and day 30 from the low-baseline (red) and high-baseline (green) groups. (d) Average IgA and IgG spider plots of 85 individuals that received 3 doses of the vaccine at day 0 (pink) and day 30 (green). Individuals were sorted by baseline response to SARS-CoV-2 VOCs. * $p < 0.05$; ** $p < 0.001$; *** $p < 0.0001$; **** $p < 0.00001$.

- Line 185-187: *are these differences expected (why are antibody levels lower in prior infected), and are they relevant as the ranges almost completely overlap?*

The finding reported is that the baseline IgA responses are higher in uninfected individuals. The sentence was not easy to follow. We have now revised it as follows:

We found that in the three and four-dose groups, the baseline IgA responses against the Wuhan RBD were significantly higher in uninfected individuals as compared to infected individuals ($p=0.042$ and $p=0.042$, **Fig. 3a**).

- Line 211-212: *Example of difficult to read result. What does the Abbott Alinity measure? It is not helpful of different terminology is used, and readers need to refer to the methods section to understand.*

We agree. We have revised the sentence as follows:

However, in the three dose group, IgG levels as measured using the Abbott Alinity assay were associated with infection risk ($HR=1.59$ $p = 0.02$), and a similar trend was observed for the Rad Bioplex S2 assay ($HR=1.39$, $p=0.089$; **Fig. 3b**, **Fig. 4c**).

REVIEWERS' COMMENTS

Reviewer #1 (Remarks to the Author):

The authors have addressed all major and minor comments . The manuscript still lacks direct mechanistic experiments . Flow cytometry with Spike specific probes or CITE seq to monitor B cell subset distribution would have added a lot to this manuscript .

Reviewer #2 (Remarks to the Author):

I commend the authors for addressing most of my concerns. I have some minor things that should be addressed.

The major comment not addressed to my satisfaction is:

- The authors find more infections in the three-dose group compared to the four dose group. However, as the third vaccine has been delivered ago, there must have been a significant difference in the time interval between previous vaccination and infection. Concretely: is the difference in infection rate a function of the number of vaccines administered, or time since last vaccination? If the authors cannot address this, it should be mentioned in the discussion as a possibility or limitation.

Additionally:

- the timing of sampling (30 days after 4th vaccination / 30 day interval) is still not clear until the results. Please clarify in the abstract.

- I am still puzzled by what 'variants' means in the figures. For example figure 1A, When and variants. Which variant was measured for that specific graph (same goes for other figures)?

- The authors could mention that they did not observe severe disease in this cohort.

Reviewer 2:

I commend the authors for addressing most of my concerns. I have some minor things that should be addressed.

The major comment not addressed to my satisfaction is:

- The authors find more infections in the three-dose group compared to the four dose group. However, as the third vaccine has been delivered ago, there must have been a significant difference in the time interval between previous vaccination and infection. Concretely: is the difference in infection rate a function of the number of vaccines administered, or time since last vaccination? If the authors cannot address this, it should be mentioned in the discussion as a possibility or limitation.

We thank the reviewer for insisting on clarifying this specific issue. It is indeed true that we cannot distinguish between the two. We have now added the following sentence to the discussion (lines 359-362):

It is also important to note that individuals who did not receive a fourth dose were a median of 177 days from their third dose. Therefore, it is possible that the protective effect of the 4th dose may be due to the shorter time interval from their last vaccination and not due to the number of vaccine doses received.

Additionally:

- the timing of sampling (30 days after 4th vaccination / 30 day interval) is still not clear until the results. Please clarify in the abstract.

We have completely revised and shortened the abstract as required.

- I am still puzzled by what 'variants' means in the figures. For example figure 1A, When and variants. Which variant was measured for that specific graph (same goes for other figures)?

We now revised the legends to include the following sentence:

a IgG and IgA magnitude to antigens from the Wuhan strain and SARS-COV-2 variants of concern including Alpha, Beta, Gamma, Delta, Iota, Kappa, Mu, Theta, and several sub-variants (see Supplementary Table 1).

In some places we simply referred to Supplementary Table 1 which lists all of the antigens used in our analysis.

- The authors could mention that they did not observe severe disease in this cohort.

We have added the following sentence: (line 105)

We did not observe any cases of severe disease in this cohort.